# The microprotein Nrs1 rewires the G1/S transcriptional machinery during nitrogen limitation in budding yeast

Sylvain Tollis[1,2]☯*, Jaspal Singh[3]☯, Roger Palou[2‡], Yogitha Thattikota[2‡¤], Ghada Ghazal[2], Jasmin Coulombe-Huntington[2], Xiaojing Tang[3], Susan Moore[2], Deborah Blake[2], Eric Bonneil[2], Catherine A. Royer[4], Pierre Thibault[2], Mike Tyers[2]*

**1** Institute of Biomedicine, University of Eastern Finland, Kuopio, Finland, **2** Institute for Research in Immunology and Cancer, University of Montréal, Montréal, Québec, Canada, **3** Lunenfeld-Tanenbaum Research Institute, Mount Sinai Hospital, Toronto, Ontario, Canada, **4** Department of Biological Sciences, Rensselaer Polytechnic Institute, Troy, New York, United States of America

☯ These authors contributed equally to this work.
¤ Current address: Montreal Neurological Institute, McGill University, Montréal, Quebec, Canada; Department of Neurology and Neurosurgery, McGill University, Montreal, Quebec, Canada
‡ RP and YT also contributed equally to this work.
* sylvain.tollis@uef.fi (ST); md.tyers@umontreal.ca (MT)

**Data Availability Statement:** Supplemental text and figures are provided in the supplemental text file S1 Text and S1–S7 Figs. Supplemental tables 1 to 6 are provided as S1–S6 Tables. Raw gels

## Abstract

Commitment to cell division at the end of G1 phase, termed Start in the budding yeast *Saccharomyces cerevisiae*, is strongly influenced by nutrient availability. To identify new dominant activators of Start that might operate under different nutrient conditions, we screened a genome-wide ORF overexpression library for genes that bypass a Start arrest caused by absence of the G1 cyclin Cln3 and the transcriptional activator Bck2. We recovered a hypothetical gene *YLR053c*, renamed *NRS1* for Nitrogen-Responsive Start regulator 1, which encodes a poorly characterized 108 amino acid microprotein. Endogenous Nrs1 was nuclear-localized, restricted to poor nitrogen conditions, induced upon TORC1 inhibition, and cell cycle-regulated with a peak at Start. *NRS1* interacted genetically with *SWI4* and *SWI6*, which encode subunits of the main G1/S transcription factor complex SBF. Correspondingly, Nrs1 physically interacted with Swi4 and Swi6 and was localized to G1/S promoter DNA. Nrs1 exhibited inherent transactivation activity, and fusion of Nrs1 to the SBF inhibitor Whi5 was sufficient to suppress other Start defects. Nrs1 appears to be a recently evolved microprotein that rewires the G1/S transcriptional machinery under poor nitrogen conditions.

## Introduction

All organisms have evolved adaptive regulatory mechanisms to optimize fitness in the face of ever-changing environmental conditions. This ability to adapt is particularly important for unicellular organisms, which lack the capacity to establish the internal homeostatic environments of metazoan species. In the budding yeast *Saccharomyces cerevisiae*, different carbon

images are provided in the supplemental file S1 Raw Images. Raw quantitative data are provided in the supplemental files S1–S7 Data as indicated in the text. The mass spectrometry data have been deposited to the ProteomeXchange Consortium via the PRIDE [Perez-Riverol Y, Csordas A, Bai J, Bernal-Llinares M, Hewapathirana S, Kundu DJ, et al. The PRIDE database and related tools and resources in 2019: improving support for quantification data. Nucleic Acids Res. 2019;47 (D1):D442-D50] partner repository with the dataset identifier PXD018681 and DOI 10.6019/ PXD018681. The RNA-sequencing data have been deposited to the Gene Expression Omnibus database (GEO, NCBI, accession number GSE179366).

**Funding:** This work was supported by a grant from the Canadian Institutes of Health Research (FDN-167277) to M.T. (https://cihr-irsc.gc.ca/), a Genomics Technology Platform award from Genome Canada to M.T. and P.T. (https://www. genomecanada.ca), a Natural Sciences and Engineering Research Council grant (NSERC 311598) to P.T. (https://www.nserc-crsng.gc.ca/), a Genomics Applications Partnership Program award from Genome Canada and Genome Quebec to P.T. (https://www.genomequebec.com/, https:// www.genomecanada.ca/), a National Science Foundation grant (PHYS 1806638) to C.A.R. (https://www.nsf.gov/), an Institute for Data Valorisation (IVADO, https://ivado.ca/) postdoctoral fellowship to J.C.-H., a Canada Research Chair in Systems and Synthetic Biology to M. T. (https:// www.chairs-chaires.gc.ca/), and a Sigrid Jusélius Foundation grant to S.T. (https://www. sigridjuselius.fi/). The funders had no role in study design, data collection and analysis, decision to publish, or preparation of the manuscript.

**Competing interests:** The authors have declared that no competing interests exist.

**Abbreviations:** ChIP, chromatin immunoprecipitation; CTD, carboxyl-terminal domain; FDR, false discovery rate; FOV, field of view; MBF, MCB-Binding Factor; MHC, major histocompatibility complex; MS–MS, tandem MS; NRS1, Nitrogen-Responsive Start regulator 1; PKA, protein kinase A; PTM, posttranslational modification; RICS, raster image correlation spectroscopy; Ribi, ribosome biogenesis; RNA-seq, RNA sequencing; RP, ribosomal protein; SBF, SCB-Binding Factor; SEP, small ORF-encoded peptide; SGA, synthetic genetic array; sN&B, scanning Number and Brightness; TOR, target of rapamycin; WT, wild-type; YNB+Pro, YNB + 0.4% proline + 2% glucose.

and nitrogen sources can dramatically affect the rates of cell growth and division, as well as developmental programs [1]. Yeast cells commit to division at the end of G1 phase, an event referred to as Start [2,3]. In order to pass Start, cells must achieve a characteristic critical size threshold that dynamically adjusts to changing nutrient availability, thereby optimizing competitive fitness [3]. How nutrient conditions modulate the growth and division machinery at the molecular level is still largely unknown.

Start initiates a complex G1/S transcriptional program of approximately 200 genes that encode proteins necessary for bud emergence, DNA replication, spindle pole body duplication, and other processes. This program is controlled by 2 transcription factor complexes, SBF (Swi4/6 Cell Cycle Box [SCB]-binding factor) and MBF (*Mlu*I Cell Cycle Box [MCB]-binding factor), each comprised of related DNA-binding proteins, Swi4 and Mbp1, respectively, coupled to a common regulatory subunit Swi6 [4,5]. Individually Swi4 and Mbp1 are not essential, but a double *swi4Δ mbp1Δ* mutant is inviable [6], consistent with the significant overlap between SBF and MBF binding sites in G1/S promoters [7–10]. In pre-Start cells that have not achieved critical cell size, SBF is inhibited by the Whi5 transcriptional repressor [11–13]. At Start, the G1 cyclin (Cln)-Cdc28 protein kinases phosphorylate both SBF and Whi5 to disrupt the SBF–Whi5 interaction and trigger Whi5 nuclear export [12–14]. The upstream G1 cyclin Cln3 is thought to initiate a positive feedback loop in which SBF-dependent expression of *CLN1/2* further amplifies Cln-Cdc28 activity and thus SBF activation [15,16]. The expression of Cln3 itself does not rely on SBF-dependent positive feedback [17]. Although *CLN3* was isolated as a potent dose-dependent activator of Start [18,19], the size-dependent mechanism whereby Cln3-Cdc28 initiates the SBF positive feedback loop remains uncertain [20–28]. *CLN3* is essential only in the absence of other parallel activators of Start, most notably *BCK2*, which encodes a general transcriptional activator [29,30].

Connections between the main yeast nutrient signaling conduits, the cell size threshold, and Start have been established. Activation of the protein kinase A (PKA) pathway by glucose represses *CLN1* and other G1/S transcripts, which may increase the cell size threshold in rich nutrients [31,32]. The TOR signaling network, which controls many aspects of cell growth including the rate of ribosome biogenesis (Ribi), has been linked to the size threshold [11,33–35]. The TORC1 complex phosphorylates and activates the effector kinase Sch9 and the master transcription factor Sfp1 to activate ribosomal protein (RP) and Ribi genes [33,34,36]. Deletion of either *SFP1* or *SCH9* abrogates the carbon source–dependent control of cell size [33]. The Rim15 kinase, which is active under respiratory growth conditions in poor carbon sources, suppresses the Cdc55 phosphatase that dephosphorylates Whi5 and thereby contributes to Whi5 inactivation even when Cln3 activity is low [37]. Poor nutrient conditions also increase the expression of the G1/S transcription factors and thereby activate Start at a smaller cell size [22]. Notably, though, a *cln3Δ bck2Δ whi5Δ* triple mutant is completely viable and still responds to nutrient cues, suggesting that nutrient regulation of Start may be partly independent of the Cln3-Bck2-Whi5-SBF axis [12,33].

To identify activators of Start that might act in parallel to the central Cln3-Bck2-Whi5 pathway, we screened for dosage suppressors of the lethal *cln3Δ bck2Δ* Start arrest phenotype [12,13,38,39]. The screen identified *YLR053c*, a poorly characterized hypothetical gene that encodes a recently evolved 108 amino acid microprotein, which we renamed *NRS1* for Nitrogen-Responsive Start regulator 1. Nrs1 was only expressed in poor nitrogen conditions and was cell cycle regulated with peak nuclear localization at Start. *Nrs1* interacted genetically and physically with SBF and caused a small size phenotype when overexpressed in wild-type cells. Nrs1 exhibited intrinsic transactivation activity and direct fusion of Nrs1 to Whi5 was sufficient to reduce cell size in rich carbon sources and rescue the *cln3Δ bck2Δ* lethality. These

results demonstrate that the recently evolved Nrs1 microprotein allows cells to adapt to poor nitrogen conditions by rewiring the Start transcriptional machinery.

## Results

### A genome-wide screen identifies overexpression of *YLR053c* as a rescue of *cln3Δ bck2Δ* lethality

We constructed a *cln3Δ bck2Δ whi5*::*GAL1-WHI5* strain that is viable in glucose but not galactose medium due to conditional *WHI5* expression [12,13]. We used synthetic genetic array (SGA) methodology to cross this query strain to an array of 5,280 strains that each of contained a *GAL1-GST-ORF* 2-μm high copy plasmid [40–42] and assessed the growth of the resulting array on galactose medium (Fig 1A and 1B). Three replicates of the screen identified 12 genes that reproducibly rescued the *cln3Δ bck2Δ* lethal Start arrest (S1 Table). These genes included the G1 cyclins *CLN1* and *CLN3* but not other known Start activators such as *CLN2*, *BCK2*, *SWI4*, and *SWI6*, possibly due to the high level of *WHI5* expression used in our screen, and/or overexpression toxicity of particular genes. We recovered 10 genes not known to be Start regulators. Notably, the short hypothetical ORF *YLR053c* restored *cln3Δ bck2Δ* growth to the same extent as *CLN3*, as validated by direct transformation of the query strain with *GAL1-YLR053c* and *GAL1-CLN3* constructs (Fig 1C). Other prospective high copy rescue genes, such as *YEA4* (Fig 1C), could not be confirmed by direct transformation.

*YLR053c* encodes a 108 residue microprotein that is only poorly characterized. Microproteins are often encoded by newly evolved proto-genes that are thought to form a genetic reservoir that fuels adaptive evolution [43]. To gain insight into *YLR053c* locus evolution, we aligned the Ylr053c protein sequence from *S. cerevisiae* with predicted orthologs from other yeast species (Fig 1D and 1E). To estimate the extent to which *YLR053c* locus evolved across these species, we used the standard dN/dS metric that measures the ratio of single DNA site substitution rates at nonsynonymous codons (dN) versus synonymous codons (dS). The *YLR053c* locus appears to have evolved relatively rapidly within the *Saccharomyces sensu stricto* group, with a dN/dS ratio in the 98th, 79th, and 90th percentile in *Saccharomyces mikatae*, *Saccharomyces bayanus*, and *Saccharomyces castellii*, respectively, as compared with *S. cerevisiae* (see S1A Fig). In comparison, the evolution of other core Start regulators was closer to the genome median, with dN/dS ratios in the 74th, 77th, and 58th percentile for *WHI5*, 69th, 75th, and 31st percentile for *SWI4*, and 82nd, 51st, and 43rd percentile for *SWI6*. A 17 amino acid sequence at the Ylr053c carboxyl-terminal region was conserved even in more distant yeasts such as *Kluyveromyces waltii* (Fig 1E). Given its genetic role at Start and expression pattern (see below), we renamed the *YLR053c* gene *NRS1* for Nitrogen-Responsive Start regulator 1.

### Nrs1 is induced by rapamycin and nitrogen limitation

To understand the function of Nrs1, we first sought to characterize its endogenous expression at the protein level. We performed scanning Number and Brightness (sN&B) confocal microscopy to localize and quantify an Nrs1-GFPmut3 fusion protein at the subcellular scale in live wild-type cells under a range of conditions [22,44]. GFPmut3, a monomeric fast-folding GFP mutant [45], will be referred to here as GFP for brevity. Nrs1-GFP was not detected in cells grown on either SC + 2% glucose or SC + 2% raffinose medium (Fig 2A), consistent with previous analysis of *YLR053/NRS1* mRNA levels [46]. In contrast, Nrs1-GFP was readily detected in the nucleus of cells grown overnight to mid log-phase in nitrogen-limited (YNB + 0.4% proline + 2% glucose, abbreviated YNB+Pro) medium (Fig 2A), also in accord with published genome-wide transcriptome analyses under various nutrient

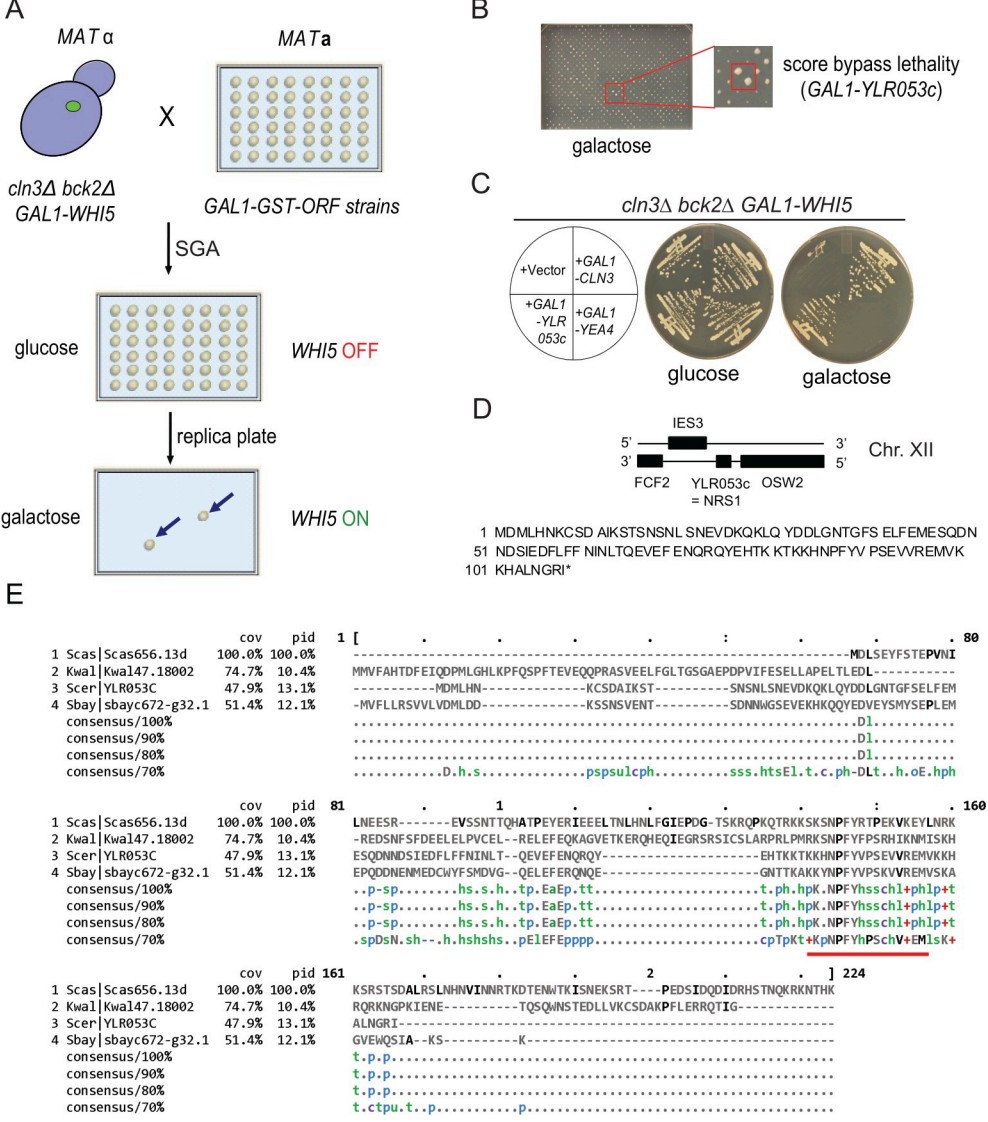

**Fig 1. A genome-wide screen identifies *YLR053c/NRS1* as a dosage suppressor of *cln3Δ bck2Δ* lethality. (A)** Schematic of SGA genetic screen for dosage suppressors of *cln3Δ bck2Δ* lethality. **(B)** Representative screen plate scored for growth on galactose medium. Red box, *GAL1-YLR053c*. **(C)** Comparison of growth at 30˚C for the indicated strains streaked onto either glucose or galactose medium. *YEA4* was a candidate hit that did not validate. **(D)** Chromosomal region around *YLR053c/NRS1* on Chr. XII and translated 108 amino acid (12.7 kDa) protein sequence. **(E)** Ylr053c/Nrs1 protein sequence in *S. cerevisiae* (top) aligned with sequences of other yeast species. Conservation of a KKXNPFYVPSXVVREMV motif at the carboxyl terminus is indicated by a red bar. *NRS1*, Nitrogen-Responsive Start regulator 1; SGA, synthetic genetic array.

conditions [47] and the presence of binding sites for the Gln3 transcription factor in the *NRS1* promoter [48,49]. Induction of Nrs1 upon the switch to nitrogen-limited YNB+Pro medium required a long incubation time, as Nrs1-GFP signal above autofluorescence background could not be detected after 7 hours in YNB+Pro but was clearly visible after 22 hours growth to mid-log phase in YNB+Pro (S1B Fig). Inhibition of TORC1 with 100 nM rapamycin induced Nrs1-GFP expression and nuclear localization within 1 hour in rich SC + 2% glucose medium (Fig 2A), as also evident by immunoblot analysis of an endogenously tagged Nrs1[13MYC] strain treated with rapamycin (S1C Fig). These expression patterns were

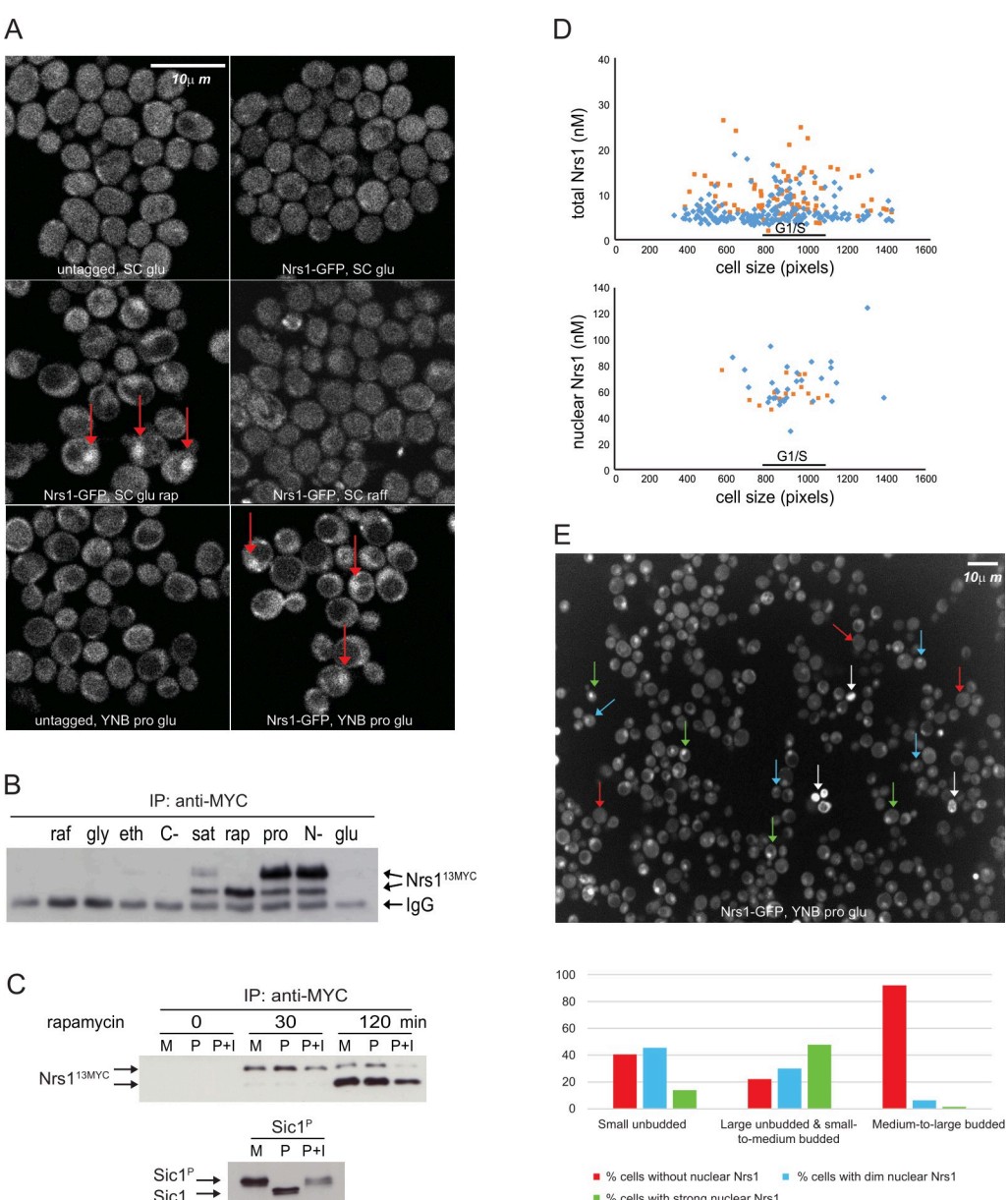

**Fig 2. Nitrogen limitation and rapamycin treatment induce cell cycle–regulated Nrs1 expression. (A)** sN&B images of untagged and *NRS1-GFP* strains. Cells were grown and imaged in SC + 2% glucose with or without 100 nM rapamycin, SC + 2% raffinose or nitrogen-limited medium (YNB +0.4% proline + 2% glucose + His, Leu, Met, Ura; labeled YNB+Pro), as indicated. Scale bar is 10 μm. The same intensity scale was used for all conditions. **(B)** Abundance of Nrs1$^{13MYC}$ in various nutrient conditions as determined by immunoblot of anti-MYC immune complexes. raf, 2% raffinose; gly, 2% glycerol; eth, 2% ethanol; C-, no carbon source; sat, saturated culture; rap, 100 nM rapamycin; pro, 0.4% proline; N-, no nitrogen source; glu, 2% glucose. IgG indicates antibody light chain. A raw image of the original immunoblot is provided in S1 Raw Images. **(C)** Nrs1$^{MYC}$ immunoprecipitates from a rapamycin treatment time course were either mock-treated (M, negative control), treated with lambda phosphatase (P), or lambda phosphatase + phosphatase inhibitors (P+I) prior to detection by anti-MYC immunoblot. *In vitro* phosphorylated recombinant Sic1 was used as a control to demonstrate activity of the phosphatase. **(D)** Absolute Nrs1 concentration in single cells grown and imaged in nitrogen-limited medium (YNB+Pro) as a function of cell size, as determined by sN&B. Blue and orange dots represent individual cells from 2 different experiments. The typical size range of cells at the G1/S transition (800 to 1,000 pixels, corresponding to 27 to 38 fL) is indicated. Cell-averaged total Nrs1 concentration (top) and nuclear concentration where Nrs1 nuclear localization was evident (bottom) are shown. Infrequent small cells with high Nrs1 levels had high autofluorescence and no nuclear localization of the signal. **(E)** Example of high-content confocal image of Nrs1-GFP cells grown to log phase in nitrogen-limited medium. Arrows indicate example cells with strong (green), dim (blue), or no nuclear Nrs1 (red) or inviable cells (white). The histogram summarizes Nrs1 signals in small unbudded

(708 cells), large unbudded and small-to-medium budded (599 cells) and medium-to-large budded cells (186 cells) from 18 confocal images. Inviable cells with high autofluorescence, out-of-focus cells for which the budding pattern could not be ascertained and regions in which illumination was not homogeneous were not scored. All numerical values underlying panels D and E may be found in S1 Data. *NRS1*, Nitrogen-Responsive Start regulator 1; sN&B, scanning Number and Brightness; YNB+Pro, YNB + 0.4% proline + 2% glucose.

confirmed with an independent fusion of Nrs1 to wild-type GFP (S1D Fig). Hyperosmotic stress, oxidative stress, and DNA damage did not induce Nrs1 expression in SC + 2% glucose medium (S1E and S1F Fig). Immunoblot analysis of a Nrs1$^{13MYC}$ strain in different carbon and nitrogen sources, as well as in rapamycin-treated and saturated cultures, confirmed the specific expression of Nrs1 only under conditions of nitrogen limitation or TORC1 inhibition (Fig 2B). A slow-migrating form of Nrs1 was induced by nitrogen limitation (Fig 2B) and also appeared first upon rapamycin treatment before conversion to the fast-migrating form (S1C Fig). To investigate the nature of this presumptive posttranslational modification (PTM) on Nrs1, we treated Nrs1$^{MYC}$ immunoprecipitates with lambda phosphatase but found that this did not affect the slower-migrating species (Fig 2C). We also did not detect Nrs1-derived phosphopeptides by mass spectrometry (see below; S4 Table). We have not been able to determine the nature of this modification on Nrs1.

## Nrs1 abundance peaks at the G1/S transition

We quantified Nrs1-GFP levels in asynchronous populations of live cells grown in nitrogen-limited YNB medium with our custom sN&B analysis software [22]. Nrs1-GFP concentrations, averaged over entire single cells, were between 5 and 20 nM. Higher levels were observed in cells close to the typical critical size at the end of G1 phase, where Nrs1-GFP was predominantly nuclear. In these cases, Nrs1 nuclear concentration was 60 to 80 nM (Fig 2D). These Nrs1 nuclear levels at Start were comparable to Swi4 (50 to 100 nM), Whi5 (100 to 120 nM), and Swi6 (130 to 170 nM) levels determined previously by the same method [22]. We further confirmed Nrs1-GFP expression and nuclear localization principally in large unbudded and small-to-medium budded cells in high-content images acquired by standard confocal microscopy (Fig 2E). The cell cycle–regulated expression pattern of Nrs1 protein suggested that it plays a role at Start, consistent with suppression of the *cln3Δ bck2Δ* arrest by *NRS1* overexpression.

## *NRS1* genetically interacts with SBF and other Start regulators

To investigate the mechanism by which Nrs1 promotes Start, we examined genetic interactions of *NRS1* with the main known Start regulators. Overexpression of *NRS1* from the *GAL1* promoter, either integrated at the *NRS1* locus or from a 2-μm high copy plasmid [50], caused a pronounced small size phenotype, in agreement with its putative role as a Start activator (Fig 3A and 3B). To interrogate the genetic requirements for this size phenotype, we transformed deletion mutants of known Start regulators with the *GAL1-NRS1* high copy plasmid and examined cell size epistasis. *NRS1* overexpression almost entirely rescued the large size phenotype of a *cln3Δ* mutant and partially rescued the large size of a *bck2Δ* mutant (Fig 3B). A *whi5Δ* mutant was epistatic to *NRS1* overexpression, whereas a *nrs1Δ* deletion did not exacerbate the larger size caused by *WHI5* overexpression and only modestly increased the size of a *whi5Δ* mutant (Fig 3C). Notably, the small size caused by *NRS1* overexpression was abrogated in *swi6Δ* and *swi4Δ* mutant strains (Fig 3D). This requirement for full SBF function was further demonstrated with a temperature-sensitive *swi4-ts* strain in which Swi4 binding to Swi6 is altered [51], grown at the semipermissive temperature of 30˚C (Fig 3E).

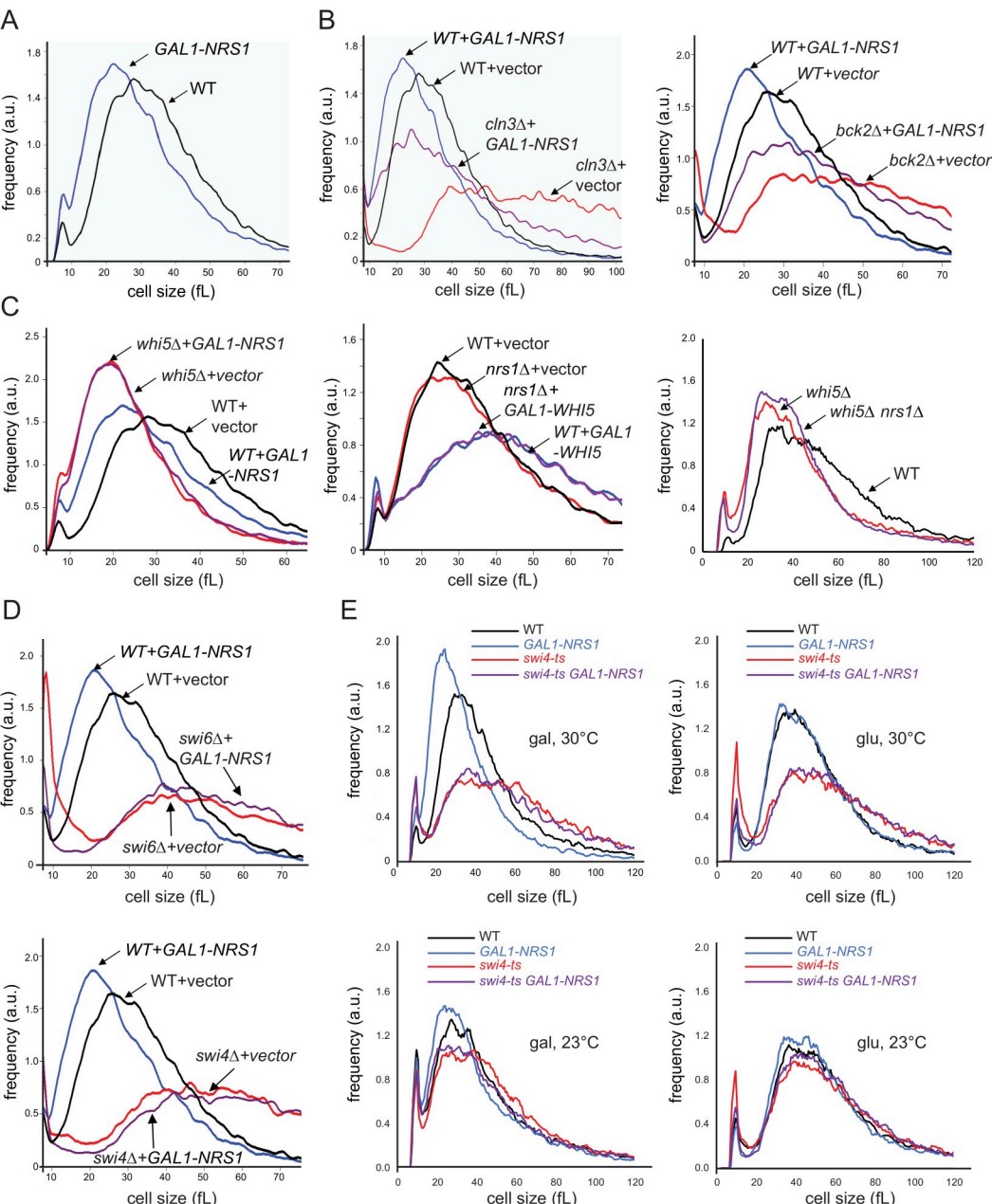

**Fig 3. _NRS1_ is genetically upstream of SBF.** Cell size distributions were determined for the indicated genetic combinations. **(A)** _NRS1_ overexpression alone. **(B)** _NRS1_ overexpression with either _cln3Δ_ or _bck2Δ_ mutations. **(C)** _NRS1_ overexpression and _nrs1Δ_ mutation with either _WHI5_ overexpression or _whi5Δ_ mutation. **(D)** _NRS1_ overexpression with either _swi4Δ_ or _swi6Δ_ mutations. **(E)** _NRS1_ overexpression with a _swi4-ts_ mutation. Strains were transformed with either _GAL1-NRS1_, _GAL1-WHI5_, or empty vector control high copy plasmids as indicated. Strains bearing galactose-regulated constructs and associated controls were induced for 6 hours in SC + 2% galactose before size determination. WT plots shown in panels B, C, and D (left, middle) represent the same measurement. All numerical values underlying this figure may be found in S2 Data. _NRS1_, Nitrogen-Responsive Start regulator 1; SBF, SCB-binding factor; WT, wild-type.

In contrast to its overexpression, deletion of _NRS1_ did not detectably affect overall growth rate or cell size in a wild-type S288C (BY4741) laboratory strain, regardless of whether cells were grown in either nitrogen-replete or nitrogen-limited medium (S2A–S2D Fig). Moreover, in competitive growth experiments, _nrs1Δ_ and wild-type cells had indistinguishable fitness

(S1 Text and S2E Fig). Because the small size phenotype caused by *NRS1* overexpression depended on *SWI4* and *SWI6*, we hypothesized that MBF might partially compensate for loss of *NRS1* function. We therefore generated an *mbp1Δ nrs1Δ* double mutant strain and evaluated growth in nitrogen-rich SC and nitrogen-poor YNB+Pro media. While the single mutant strains had no growth defect in either condition, the *mbp1Δ nrs1Δ* double mutant had a pronounced growth defect that was specific to nitrogen-poor conditions (Fig 4A). We furthermore considered the possibility that laboratory strains may have lost some requirements for *NRS1* through the course of decades-long propagation in artificially rich nutrient conditions. To test this idea, we deleted *NRS1* in the prototrophic wild yeast *Saccharomyces boulardii*, which shares >99% of its genome with *S. cerevisiae* including *NRS1* [52]. An *S. boulardii* strain lacking *NRS1* grew slower than wild-type cells in nitrogen-poor medium specifically (i.e., YNB+Pro+0.1% glucose medium not supplemented with amino acids, Fig 4B). Hence, under conditions of nitrogen limitation, *NRS1* promotes growth in genetically crippled contexts in laboratory strains and is required for optimal growth of a wild variant of *S. cerevisiae*. Taken together, these genetic interactions suggested that Nrs1 promotes the G1/S transition by acting upstream of the Whi5-inhibited form of SBF in a manner that is independent of Cln3 and Bck2.

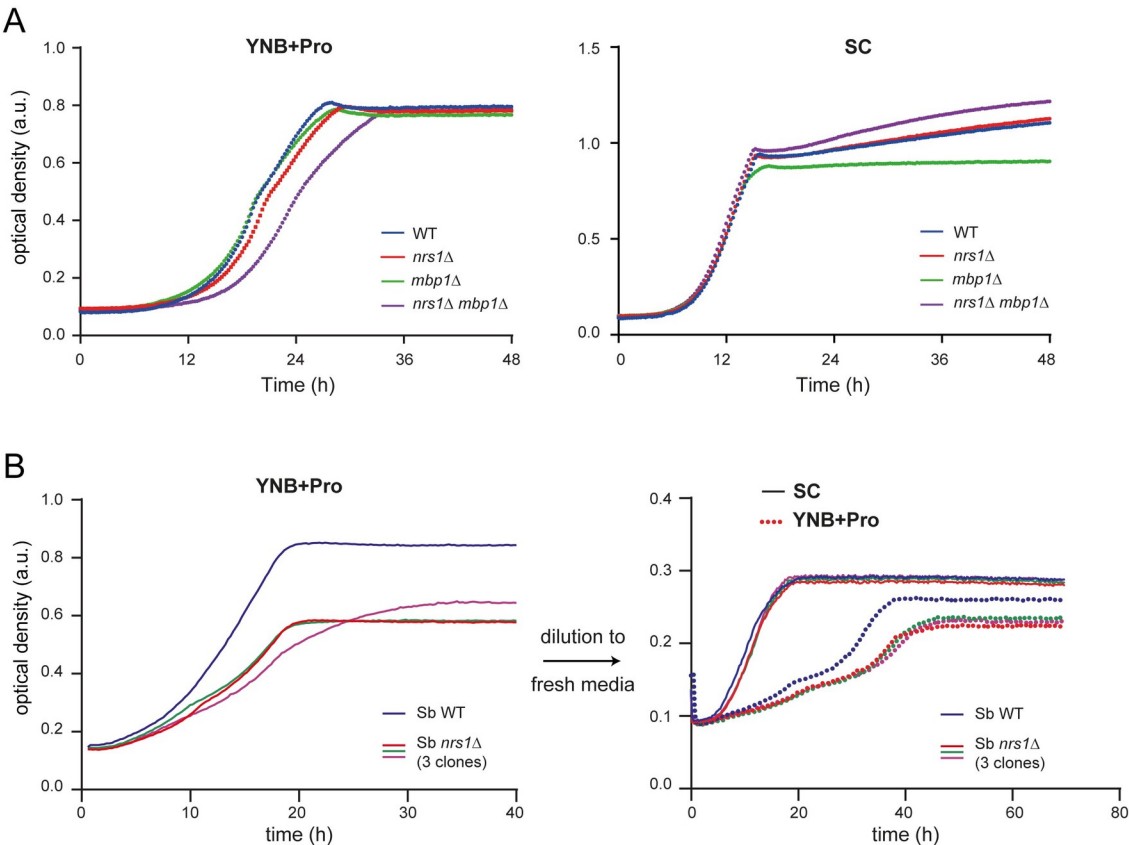

**Fig 4. *NRS1* is required for optimal growth in poor nitrogen medium.** (**A**) Growth curves for WT (BY4741), *nrs1Δ*, *mbp1Δ*, and *mbp1Δ nrs1Δ* laboratory strains grown in nitrogen-limited YNB+Pro+0.1% glucose medium and nitrogen-rich SC+0.1% glucose medium. Curves represent the average of 3 different clones for each mutant strain. (**B**) Growth curves for WT and *nrs1Δ S. boulardii* strains were grown in either nitrogen-poor YNB+Pro+0.1% glucose minimal medium (i.e., not supplemented with any other amino acids) or nitrogen-rich SC+0.1% glucose medium. Three different isolates for the *nrs1Δ S. boulardii* strain were analyzed. Growth curves are the average of at least 12 independent colonies for each strain. To eliminate potential nutrient lag effects, cultures were rediluted into fresh medium, and their growth was monitored again. All numerical values are provided in S3 Data. *NRS1*, Nitrogen-Responsive Start regulator 1; WT, wild-type; YNB+Pro, YNB + 0.4% proline + 2% glucose.

## Nrs1 interacts with SBF *in vivo* and *in vitro*

We next sought to identify Nrs1 protein interactors under conditions in which Nrs1 was endogenously expressed at physiological levels. We performed immunoaffinity purification on lysates from an endogenously tagged *NRS1*[13MYC] strain and a control untagged strain grown in nitrogen-limited YNB+Pro medium. Analysis of the samples by mass spectrometry (see Methods and S1 Text) identified multiple peptides (S4 Table) from which the corresponding proteins were identified (>2 unique peptides, false discovery rate [FDR] <1%, S5 Table). Proteins specific to the Nrs1[13MYC] sample were identified by subtracting proteins present in the untagged control and filtering hits against the CRAPome database of nonspecific interactions [53]. This workflow yielded 7 proteins that were specific to the Nrs1[13MYC] sample (S3 Table and S3A Fig). Of these, Nrs1 was represented by 7 peptides (48% coverage), Swi4 by 5 peptides, and Swi6 by 5 peptides; the remaining 4 candidates were represented by only 2 or 3 peptides and thus of lower confidence. This unbiased analysis demonstrated that endogenous Nsr1 expressed under physiological conditions interacted with SBF. To confirm this result, we performed co-immunoprecipitation experiments on extracts of rapamycin-treated cells expressing either *SWI4*[3FLAG] or *SWI6*[3FLAG] alleles in combination with either *NRS1*[13MYC] or *WHI5*[13MYC] alleles, each expressed from their respective endogenous promoter. Swi4[3FLAG] and Swi6[3FLAG] were each immunoprecipitated with anti-FLAG resin and then blotted with antibodies against either FLAG or MYC epitope tags. Nrs1[13MYC] and Whi5[13MYC] were detected in immunoprecipitates of endogenous Swi4[3FLAG] and Swi6[3FLAG] from cells grown the presence of rapamycin but not in control immunoprecipitates from strains that lacked the FLAG-tagged alleles (Figs 5A and S3B). A similar experiment carried out with polyclonal anti-Swi4 and anti-Swi6 antibodies specifically detected endogenous SBF in Nrs1[13MYC] immunoprecipitates from cells grown in the presence but not the absence of rapamycin (S3C Fig). Collectively, these results suggested that Nrs1 directly interacted with SBF under physiological conditions.

To determine if the interaction between Nrs1 and SBF was direct, we carried out *in vitro* binding assays with purified recombinant proteins. We titrated recombinant [FLAG]Swi4-Swi6 and [FLAG]Mbp1-Swi6 complexes produced in baculovirus-infected insect cells [12] against recombinant [GST]Nrs1 produced in *E. coli* and immobilized on GSH-Sepharose resin. Swi4 and Swi6 were both captured with [GST]Nrs1 across the titration series, whereas control GSH-Sepharose resin did not bind SBF (Fig 5B). In contrast, [GST]Nrs1 did not detectably interact with the [FLAG]Mbp1-Swi6 complex under the conditions tested, suggesting that specificity for Nrs1 is determined by the DNA-binding subunit and not the common Swi6 subunit of SBF/MBF. To test whether Nrs1 was able to interact with Whi5-bound SBF, we titrated purified recombinant [GST]Nrs1 into a preformed recombinant [GST]Whi5-[FLAG]Swi4-Swi6 complex. We observed that Nrs1 bound effectively even in the presence of Whi5 (Fig 5C). These results demonstrated that endogenous Nrs1 interacts directly and specifically with the Swi4-Swi6 complex, that no additional factors are needed for this interaction to occur, and that Nrs1 can bind SBF even in the presence of Whi5.

## Nrs1 binds to SBF-regulated promoters *in vivo*

We next assessed the presence of Nrs1[13MYC] at the SBF-regulated promoters of *CLN2* and *PCL1* by chromatin immunoprecipitation (ChIP). For both genes, we specifically detected SCB-containing promoter sequences by PCR in cross-linked anti-MYC immune complexes purified from cells that expressed Nrs1[13MYC] from the endogenous *NRS1* locus (Fig 6A). The enrichment of *PCL1* and *CLN2* sequences in Nrs1[13MYC] immunoprecipitates only in rapamycin-treated cells further demonstrated that Nrs1 specifically binds SBF-regulated G1/S promoter DNA (Fig 6A).

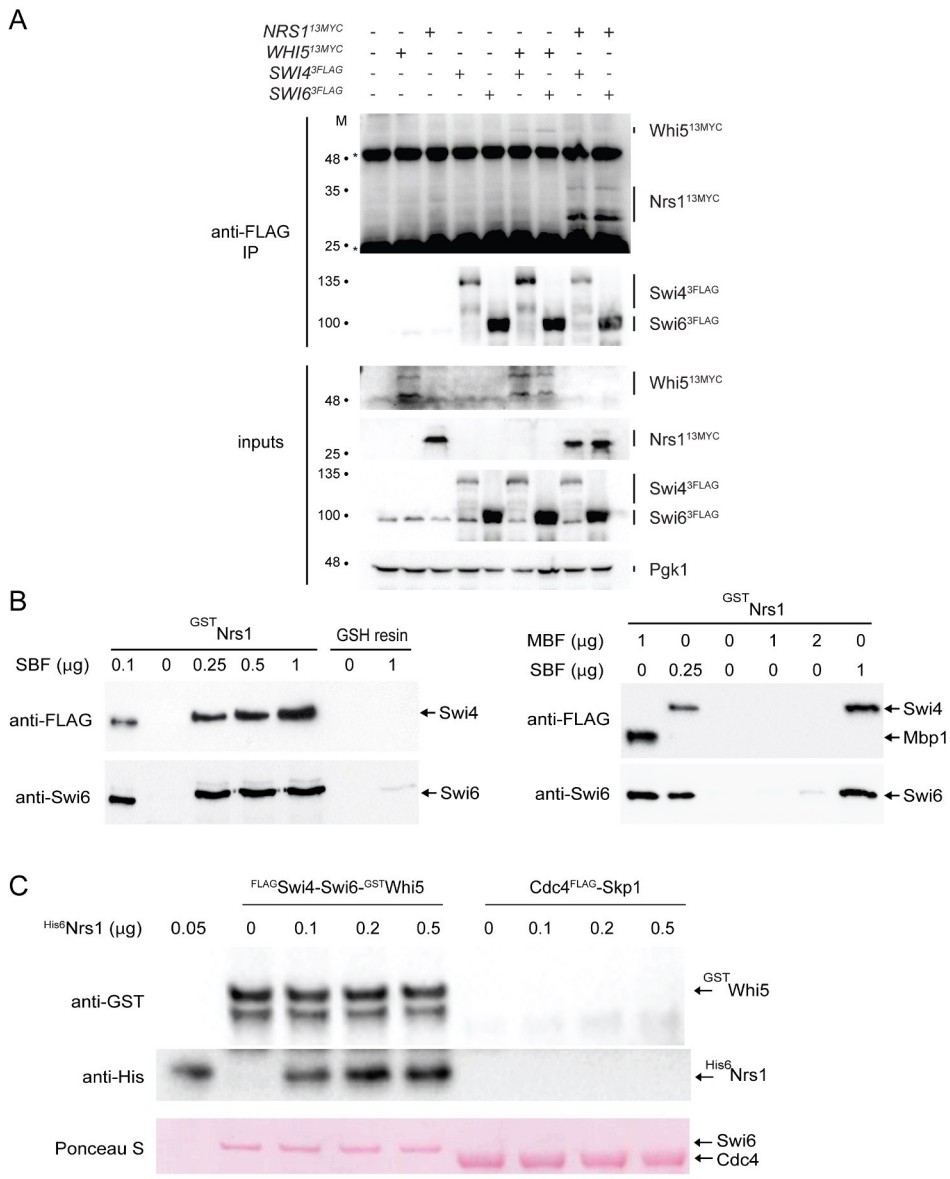

**Fig 5. Nrs1 binds to SBF *in vivo* and *in vitro*. (A)** Swi4³FLAG or Swi6³FLAG complexes were immunoprecipitated from the indicated strains grown in the presence of 200 nM rapamycin for 2 hours and interacting proteins assessed by immunoblot with the antibodies against the indicated tags or proteins. Co-immunoprecipitation of Whi5¹³MYC with Swi4³FLAG and Swi6³FLAG served as a positive control. Mr markers (M) are indicated for each blot. Pgk1 served as a loading control. Note the lower Mr form of Whi5 visible in the input blot is obscured by the IgG heavy chain signal in the immunoprecipitations. Data is representative of at least 5 independent experiments (see S3B Fig for a replicate experiment and S3C Fig for a reciprocal Nrs1¹³MYC and Whi5¹³MYC co-immunoprecipitation experiment performed with polyclonal anti-Swi4 and anti-Swi6 antibodies). **(B)** Recombinant ᴳˢᵀNrs1 immobilized on GSH-Sepharose resin was incubated with increasing concentrations of soluble purified SBF (ᶠᴸᴬᴳSwi4-Swi6) or MBF (ᶠᴸᴬᴳMbp1-Swi6). Bound proteins were analyzed by immunoblot with anti-Swi6 and anti-FLAG antibodies as indicated. GSH-Sepharose resin alone served as a negative control. **(C)** Recombinant ᶠᴸᴬᴳSwi4-Swi6-ᴳˢᵀWhi5 complexes containing 0.1 ug of ᴳˢᵀWhi5 were incubated with increasing amounts of recombinant His₆-tagged Nrs1 (0.1 ug, 0.2 ug, and 0. 5ug, equivalent to 4, 8, and 20 Nrs1:Whi5 molar ratio), immunoprecipitated using anti-FLAG beads to capture SBF–Whi5 complexes and probed with anti-GST antibody to detect Whi5 and anti-HIS₆ antibody to detect Nrs1. Co-immunoprecipitation with an irrelevant recombinant protein complex (Cdc4-ᶠᴸᴬᴳSkp1) on anti-FLAG beads served as a negative control for interaction specificity. Ponceau S stain was used to demonstrate equivalent input protein complexes in each lane. Raw images of all original immunoblots are provided in the S1 Raw Images. MBF, MCB-binding factor; *NRS1*, Nitrogen-Responsive Start regulator 1; SBF, SCB-binding factor.

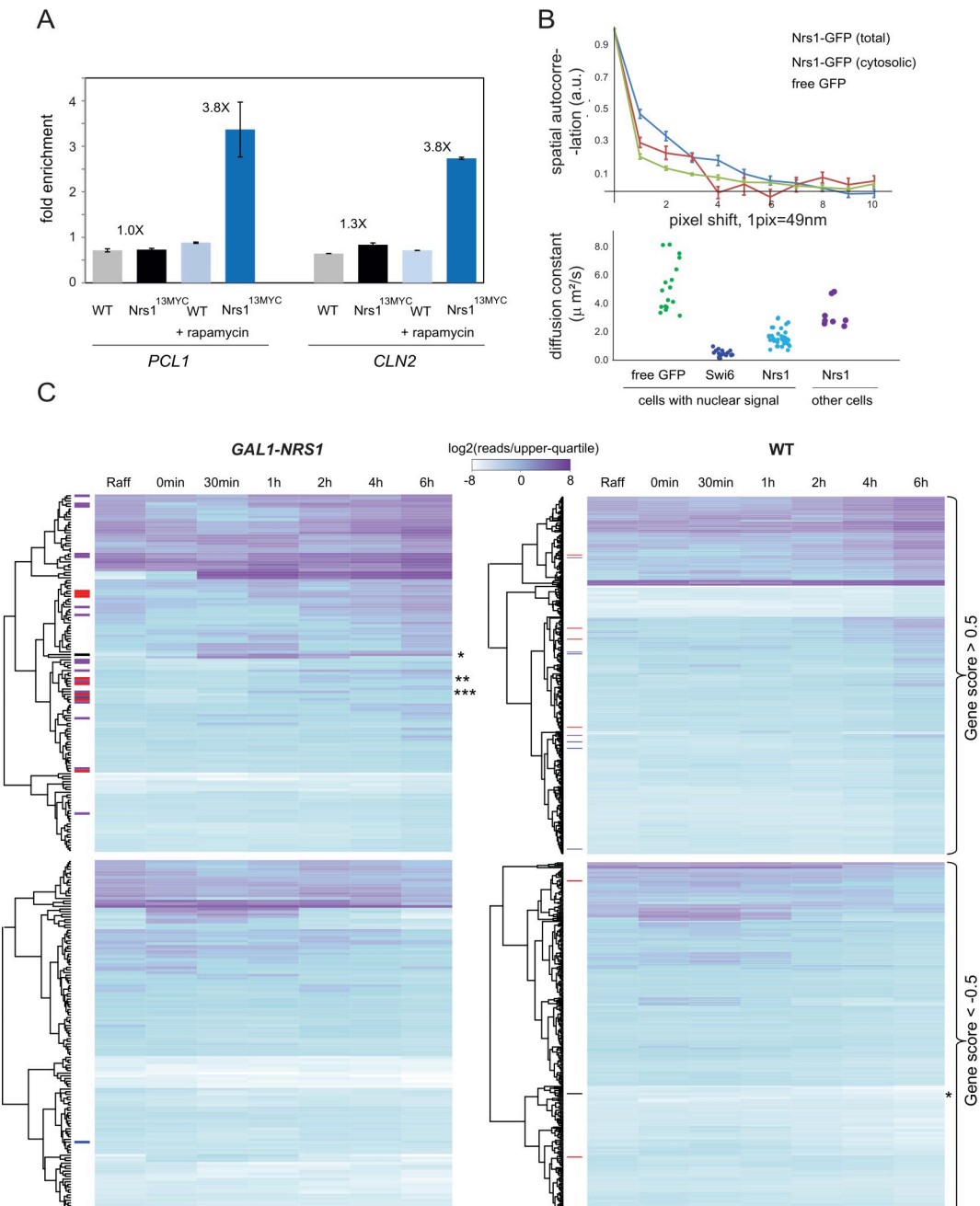

**Fig 6. Nrs1 binds to SBF-regulated promoter DNA and activates SBF-driven transcription. (A)** Anti-MYC chromatin immunoprecipitates from WT untagged control and *NRS1*[13MYC] strains either untreated or treated with 200 ng/mL rapamycin for 2 hours were probed for the presence of *CLN2* and *PCL1* promoter DNA sequences by real-time quantitative PCR. Bars indicate the mean fold-enrichment across 2 replicates, and error bars show the standard error on the mean. **(B)** Top panel: RICS vertical correlations from a *NRS1-GFP* strain grown in nitrogen-limited medium (YNB+Pro) as a function of the pixel shift for total and cytosolic Nrs1-GFP pools. RICS correlation for free GFP was used as a control for unconstrained diffusion. Data points show correlation averages over N FOV ($N = 25$ for nuclear Nrs1, $N = 7$ for cytosolic Nrs1, $N = 12$ for free GFP). Each FOV contained 1 to 5 cells. Error bars represent the standard error on the mean. Bottom panel: Scatter plots of fitted diffusion coefficients from individual FOVs of the same strains. Data from Swi6-GFP FOVs served as a control for constrained diffusion. All numerical values underlying panels A and B may be found in S5 Data. **(C)** Heatmaps of gene expression profiles upon galactose induction of *GAL1-NRS1* and WT control strains. Genes were selected based on a gene rank score across biological triplicates and values shown are upper quartile-normalized log₂-read counts averaged across triplicates for each time point. Genes that were strongly up-regulated (gene score > 0.5) and down-regulated (gene score < −0.5) are shown. Genes were clustered for *GAL1-NRS1* and WT heatmaps separately according to the default

settings of the R Heatmap function. Positions of NRS1 (\*), CLN2 (\*\*), and PCL1 (\*\*\*) are indicated. SBF (red), MBF (blue), and SBF/MBF (purple) target genes are indicated on the left side. Raw data underlying this panel may be found on the Gene Expression Omnibus database under the accession number GSE179366. FOV, field of view; MBF, MCB-binding factor; *NRS1*, Nitrogen-Responsive Start regulator 1; RICS, raster image correlation spectroscopy; SBF, SCB-binding factor; WT, wild-type; YNB+Pro, YNB + 0.4% proline + 2% glucose.

We predicted that Nrs1 binding to chromatin should reduce Nrs1 mobility in the nucleus. To test this hypothesis, we assessed the molecular dynamics of Nrs1-GFP in cells grown in nitrogen-limited medium using raster image correlation spectroscopy (RICS). The RICS method exploits the hidden time structure of imaging scans, in which adjacent pixels are imaged a few microseconds apart, to quantify the diffusional properties of fluorescent molecules [54–56]. Intensity correlations between pixels shifted along the direction of scanning (horizontal correlations), and along the orthogonal direction (vertical correlations), are averaged over multiple scans. Horizontal and vertical correlations that decay with increasing pixel shift are characteristic of the dynamical properties on 10 to 100 μs and 5 to 50 ms timescales, respectively. We found that the vertical RICS correlations of nuclear Nrs1-GFP signal decreased on a slower timescale than either free nuclear GFP or cytosolic Nrs1-GFP signal (Fig 6B). Fitted Nrs1-GFP diffusion coefficients in cells in which Nrs1 was predominantly nuclear were lower compared to cells in which it was mostly cytosolic and also lower than the diffusion coefficient of free nuclear GFP, a proxy for free diffusion in the nucleus. The Nrs1 diffusion coefficient was similar to that of Swi6-GFP which is largely DNA bound (Fig 6B). This result indicated that Nrs1 associates with a slow-moving nuclear component, most likely chromatin.

The apparent size epistasis between *whi5Δ* and *GAL1-NRS1* prompted us to ask whether Nrs1 might reduce the association Whi5 with SBF and/or chromatin. However, ChIP analysis of Whi5HA at the *CLN2* and *PCL1* promoters revealed that Whi5HA–promoter interactions were not reduced in the presence of overexpressed *NRS1* (S4A Fig). In addition, *GAL1-NRS1* cells had wild-type levels of Whi5 as determined by sN&B quantification (S4B Fig), and *GAL1-NRS1* did not alter Whi5HA association with Swi4 or Swi6 in co-immunoprecipitation experiments (S4C Fig). Nrs1 and Whi5 also did not compete for SBF binding in *in vitro* binding assays with recombinant proteins (S4D Fig). Together, these results obtained using different *in vivo*, *in vitro*, biochemical, and imaging-based approaches suggested that Nrs1 binds SBF at G1/S promoters *in vivo*, but that the binding of Nrs1 does not alter Whi5 interactions with SBF or promoter DNA. These results led us to consider the possibility that Nrs1 might directly activate transcription.

## Ectopic expression of *NRS1* activates the G1/S regulon

To assess the potential role of Nrs1 in transcription, we used RNA sequencing (RNA-seq) to determine the genome-wide transcriptional response to ectopic *NRS1* overexpression. Three biological replicates of *nrs1::GAL1-NRS1* and wild-type cells were grown to log-phase in SC +2% raffinose medium, induced with galactose for 6 hours, and transcriptional profiles analyzed by RNA-seq. Gene scores were defined based on their expression fold-change before and after 6-hour induction, comparing for each gene the fold-change across multiple replicates to control for intersample expression variability and the effects of galactose (see Methods; all scores provided in S6 Table). Up-regulated genes in the *GAL1-NRS1* samples included *NRS1* itself as expected (rank 1) and many of the 139 SBF/MBF target genes that comprise the G1/S regulon as defined by integration of multiple microarray-based analyses of G1/S regulated genes [10]. These G1/S genes included *CLN2* (rank 12) and *PCL1* (rank 82), the promoters of which were bound by Nrs1 as shown above. Nrs1 also strongly induced the expression of *HO*

(rank 10), *RNR1* (rank 15), and *HCM1* (rank 20), which are markers of the G1/S transition. Visualization of the data in a heatmap format illustrated the overall enrichment of SBF/MBF target genes (Fig 6C). Among the most strongly up-regulated genes (score >0.5), SBF targets were enriched 12-fold compared to random expectation (24 of 86 genes, p = $3^*10^{-20}$), MBF targets were enriched 7-fold (17 of 101 genes, $p = 9^*10^{-11}$), but out of the 17 strongly up-regulated MBF targets, 16 were also SBF targets so that MBF-only targets were not enriched (1/53 genes, $p = 0.65$). In the wild-type control samples, no significant enrichment/depletion compared to random expectations was observed. We note that in addition to recognition of its cognate SCB elements, SBF can also activate genes that only contain MCB elements [57] such that the apparent specificity of Nrs1 for SBF is not undermined by the up-regulation of MBF genes in these experiments.

## Nrs1 confers transcriptional activity that can rescue G1/S-transcription deficient mutants

We next asked whether Nrs1 might itself function directly as a transcriptional activator. To test this hypothesis, we constitutively expressed a Nrs1-Gal4 DNA-binding domain fusion protein (*GAL4^{DBD}-NRS1*) in a reporter strain bearing the *HIS3* gene under control of the *GAL1* promoter. We used *GAL4^{DBD}* alone and *GAL4^{DBD}* fused to an irrelevant human gene (*UBE2G2*) that potently transactivates as negative and positive controls, respectively. Full-length *NRS1*, but not a truncated version that encoded only the conserved carboxyl terminus, was able to activate transcription of the *HIS3* reporter and thereby allow cell growth in medium lacking histidine (Figs 7A and S5A).

Given the inherent transactivation capacity of Nrs1, we next investigated whether *NRS1* overexpression could suppress the growth defects caused by loss of other G1/S activators. We first tested a *cln1Δ cln2Δ cln3Δ MET-CLN2* strain, in which G1 cyclin activity is restricted to methionine-free media, with or without a <*pGAL1-NRS1*> plasmid. Repression of *CLN2* by methionine caused a growth arrest, whether or not *NRS1* was expressed, although ectopic expression of *NRS1* did modestly reduce cell size and slightly increased cell proliferation (S5B Fig). Hence, *NRS1* cannot compensate for the total lack of G1-cyclin activity. We then tested whether *NRS1* overexpression could suppress the growth defects of mutants in which G1/S transcription was impaired, but in which Cln-Cdc28 activity was preserved. We first tested a *mbp1Δ swi4-ts* double mutant [58]. The growth of a *GAL1-NRS1 mbp1Δ swi4-ts* strain was improved with respect to the *mbp1Δ swi4-ts* control at a semipermissive temperature of 30° in galactose medium (Fig 7B; see S5C Fig for plate images with a 2-day extension of the incubation period), but not in other media or at the fully permissive temperature (S5B Fig). As a further test, we examined *NRS1*-mediated suppression in strains that expressed *WHI5^{12A}* and *SWI6^{SA4}* alleles in which all Cdc28 consensus sites are mutated, a combination that abrogates the Cln-Cdc28 mediated relief of SBF inhibition by Whi5 [12,14]. Overexpression of *NRS1* also partially rescued the growth defect of this strain (Fig 7C). These results suggested that Nrs1 provides an alternative mechanism of Start activation by augmenting G1/S transcription in the presence of Whi5-mediated inhibition.

## Nrs1 can bypass Whi5 inhibition of SBF

To test whether physiological levels of Nrs1 targeted to SBF are sufficient to promote Start activation, we constructed a strain expressing a Whi5-Nrs1-GFP fusion protein from the endogenous *WHI5* promoter. Experiments with this strain were carried out in SC complete medium to avoid other possible nitrogen-dependent parallel inputs to Start and to focus on Nrs1 function. The chimeric protein was expressed and had the expected molecular mass (S6A Fig) and

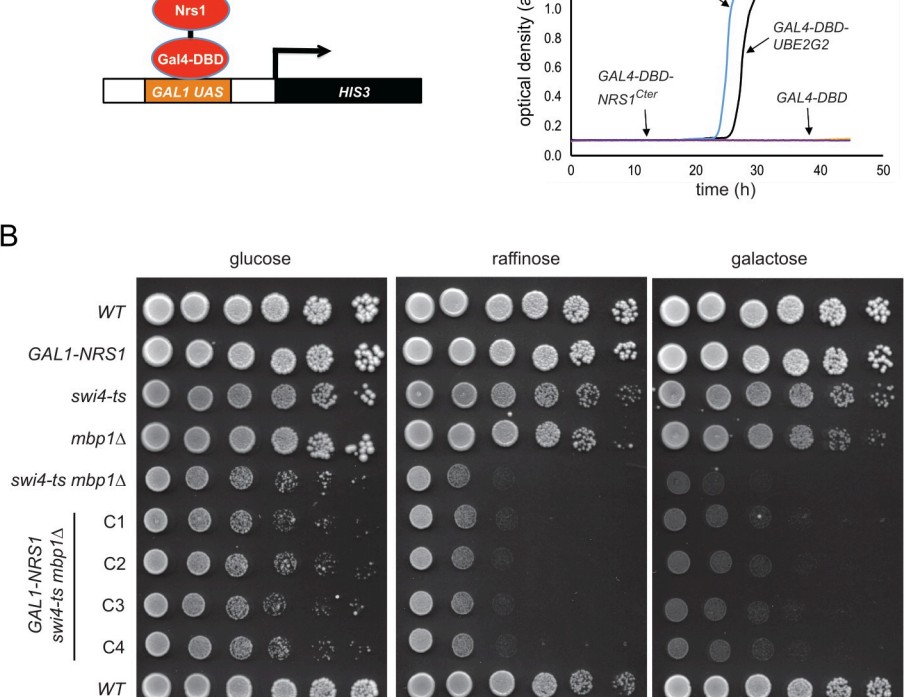

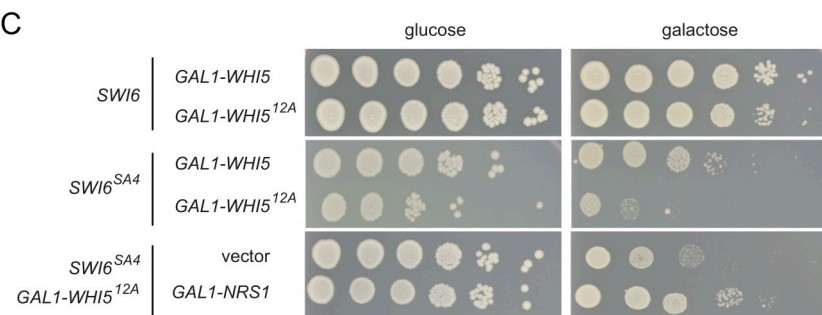

**Fig 7. Nrs1 has an inherent transcription activation function that can partially rescue G1/S-transcription deficient mutants. (A)** Transactivation of a *HIS3* reporter by fusion of Nrs1 to the Gal4 DNA-binding domain (*GAL4$^{DBD}$-NRS1*) but not by fusion of the Nrs1 carboxyl terminus (*GAL4$^{DBD}$-NRS1$^{Cter}$*). *GAL4$^{DBD}$-UBE2G2* and *GAL4$^{DBD}$* constructs served as positive and negative controls, respectively. Growth curves were determined in -His -Trp medium. All numerical values underlying this panel may be found in S6 Data. **(B)** Serial 5-fold dilutions of *NRS1* and *GAL1-NRS1* strains in WT, *swi4-ts*, *mbp1Δ*, and *mbp1Δswi4-ts* backgrounds were spotted onto SC + 2% glucose, SC + 2% raffinose and SC + 2% galactose, and grown for 5 days at 30°C. C1 to C4 indicate 4 independent clones of the *mbp1Δ swi4-ts GAL1-NRS1* strain. **(C)** Serial 5-fold dilutions of WT or *swi6Δ* strains transformed with *GAL1-WHI5*, *GAL1-WHI5$^{12A}$*, *GAL1-SWI6$^{SA4}$*, *GAL1-NRS1*, or empty control plasmids or combinations thereof were spotted on rich medium containing 2% glucose or 2% galactose and grown for 2 days at 30°C. NRS1, Nitrogen-Responsive Start regulator 1; WT, wild-type.

retained the localization pattern of endogenous Whi5, with about 30% to 35% of pre-Start cells showing a prominent nuclear signal (Figs 8A and S6B). Cells expressing Whi5-Nrs1-GFP were almost as small as *whi5Δ* cells (Fig 8B). This phenotype was not due to the presence of the GFP tag since size distributions of strains expressing untagged versus GFP-tagged Whi5 were

indistinguishable (S6C Fig). Moreover, cells expressing the chimeric protein grew at the same rate as wild-type and *whi5Δ* strains (S6D Fig). Strikingly, cells expressing Whi5-Nrs1-GFP were as small as cells that expressed *GAL1-NRS1* in galactose medium (Fig 8C).

As a reduction of Whi5 dosage in the Whi5-Nrs1-GFP fusion context could in principle explain the small size of this strain, we used sN&B to compare the nuclear concentrations of Whi5 in the wild-type and the Nrs1-fusion contexts. Whi5 levels in wild-type cells were 110 to 130 nM, consistent with those determined previously [22]. The Whi5-Nrs1-GFP chimera was present at a slightly lower concentration, 80 to 100nM (Fig 8A), close to endogenous Nrs1 nuclear levels in nitrogen-limited media (60 to 80 nM, see Fig 2D). This slight decrease in Whi5 abundance was unlikely to explain the small size of cells expressing Whi5-Nrs1-GFP since cell size is not strongly sensitive to *WHI5* gene dosage [25,59]. We confirmed that hemizygous *WHI5/whi5Δ* diploid cells were only marginally smaller than wild-type diploid cells (S6E Fig) and, moreover, *WHI5* overexpression only causes a 20% to 30% increase in mode size (Fig 3C), in agreement with previous results [25,59]. Consistently, our previously published Start model [22] also predicted that down-regulating Whi5 levels from 120nM to 85nM should not affect the critical size at Start (S6F Fig).

Finally, we predicted that the Whi5-Nrs1 fusion should be sufficient to rescue the lethal Start arrest of a *cln3Δ bck2Δ* strain. We crossed a *cln3Δ* strain bearing the integrated *WHI5-NRS1-GFP* allele to a *bck2Δ* strain and analyzed growth of dissected tetrads on selective media to identify spore genotypes (Fig 8D). We recovered many viable *cln3Δ bck2Δ WHI5-NRS1-GFP* triple mutant spore clones but did not recover any viable *cln3Δ bck2Δ WHI5* double mutant clones (Fig 8D). As expected, no viable *cln3Δ bck2Δ* spore clones were obtained from a control cross of a *cln3Δ whi5::WHI5-GFP* and a *bck2Δ* strain (Fig 8D). As a further control to ensure that the Nrs1 fusion did not merely inactivate Whi5 in a nonspecific manner, we fused the above transcriptionally inactive carboxyl-terminal fragment of Nrs1 to Whi5 and showed that it did not reduce cell size (S7A Fig) nor rescue the *cln3Δ bck2Δ* lethality (S7B Fig). These results supported a model in which Nrs1 bypasses Whi5 inhibition by directly conferring transactivation activity on the Whi5-inhibited SBF complex.

## Discussion

On the premise that additional genes may activate Start under suboptimal nutrient conditions, we screened for genes that can circumvent the Start arrest caused by loss of both *CLN3* and *BCK2* function. We discovered that *NRS1* overexpression efficiently bypasses the lethality of a *cln3Δ bck2Δ* strain and activates Start in wild-type cells; that Nrs1 associates with SBF at G1/S promoter DNA and promotes SBF-dependent transcription; that *NRS1* overexpression can genetically suppress defects in SBF function; and that Nrs1 is itself a transcriptional activator. Our data suggest a model whereby nitrogen limitation or TORC1 inhibition promote *NRS1* expression and that Nrs1 binding to SBF directly recruits the transcriptional machinery to overcome Whi5-mediated inhibition of SBF (Fig 8E). In effect, endogenous Nrs1 acts as a nutrient-dependent parallel input into SBF. Although this input appears dispensable under most growth conditions, it is revealed in a genetically sensitized context in laboratory strains and in wild yeast strains. We stress that the *cln3Δ bckΔ GAL1-WHI5* parental background used for the original high copy suppression screen might have biased towards the discovery of Start activators able to operate in presence of high Whi5 dosage.

This model of Nrs1 function (Fig 8E) allows reinterpretation of some previous *YLR053c/ NRS1* genetic interactions uncovered in high-throughput studies [60]. Deletion of *NRS1* slightly aggravates the growth defect of a *ESS1* prolyl isomerase mutation which, in turn, exhibits negative genetic interactions with deletions of *SWI6* or *SWI4* [61]. Although the latter

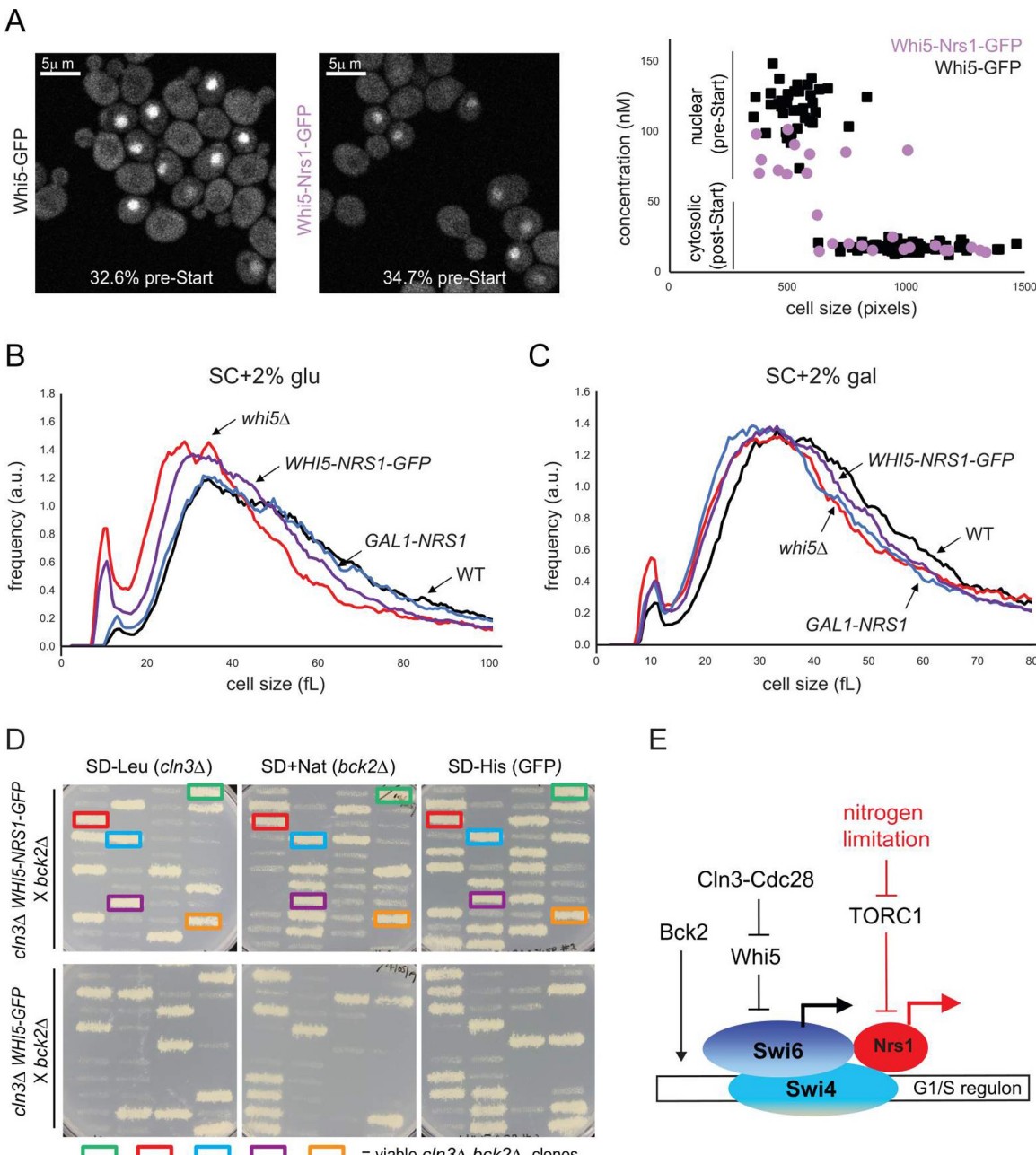

**Fig 8. Tethered Nrs1 bypasses the Whi5-dependent lethal arrest of a *cln3Δ bck2Δ* strain. (A)** sN&B microscopy images of *WHI5-GFP* and *WHI5-NRS1-GFP* cells grown in SC + 2% glucose. Scale bars are 5 μm. The same intensity scale was used on both images. Fractions of pre-Start G1 cells were computed based on the assessment of nuclear GFP signal in *WHI5-GFP* cells (*N* = 245) and *WHI5-NRS1-GFP* (*N* = 95) cells. Absolute concentrations of Whi5-GFP and Whi5-Nrs1-GFP are shown in the plot. **(B)** Cell size distributions of WT, *GAL1-NRS1*, *whi5Δ*, and *WHI5-NRS1-GFP* strains in SC + 2% glucose. **(C)** Cell size distributions of WT, *GAL1-NRS1*, *whi5Δ*, and *WHI5-NRS1-GFP* strains in SC + 2% galactose. All numerical values underlying panels A–C may be found in S7 Data. **(D)** Genotype of 10 tetrads from a *cln3Δ WHI5-NRS1-GFP* X *bck2Δ* cross (top) and a *cln3Δ WHI5-GFP* X *bck2Δ* cross (bottom). For each tetrad, spore clone growth was assessed on SD-LEU (indicates *cln3::LEU2*), SC+NAT (indicates *bck2::NAT^R*), and SD-HIS (indicates *whi5:: WHI5-NRS1-GFP-HIS3* or *whi5::WHI5-GFP-HIS3*). Colored boxes indicate viable *cln3Δ bck2Δ* spore clones, all of which also contained the *WHI5-NRS1-GFP* construct. **(F)** Simplified schematic for Nrs1-dependent activation of Start. Red lines indicate nitrogen-limited conditions. Not all components of the Start machinery are shown. See text for details. *NRS1*, Nitrogen-Responsive Start regulator 1; sN&B, scanning Number and Brightness; WT, wild-type.

interactions suggested a possible role of Ess1 in Swi6/Whi5 nuclear import, we did not observe mis-localization of Swi4, Swi6 or Whi5 in either *GAL1-NRS1* or *nrs1Δ* strains. With respect to interactions with the transcriptional machinery, *NRS1* overexpression exacerbates the defect caused by deletion of *CTK1*, the catalytic subunit of RNA Pol II carboxyl-terminal domain (CTD) kinase I [62]. In contrast, *NRS1* deletion shows a positive genetic interaction with the RNA Pol II CTD-associated phosphatase, FCP1, which negatively regulates transcription [63]. *NRS1* overexpression also subtly increases chromosomal instability [64], which might result from premature Start activation [65]. These interactions with transcriptional regulators are consistent with our proposed model for Nrs1 function in G1/S transcription activation.

   *NRS1* is a member of a newly described class of genetic elements variously called proto-genes, neo-ORFs, smORFS, small ORF encoded peptides (SEPs), or microproteins [43,66–74]. Computational predictions and ribosomal footprinting first identified hundreds of short species-specific translated peptides from extragenic regions in yeast [43]. Subsequent approaches have yielded a plethora of microproteins encoded by smORFs in various species [66–69,75]. Ribosome profiling and high-throughput genetic analyses has revealed hundreds of functional microproteins in human cells [76,77]. Documented functions for microproteins include phagocytosis [70], autophagy [71], actin-based motility [72], mRNA decapping [78], proteolytic processing [79], mitotic chromosome segregation [73], mitochondrial morphology control [76], plant hormone signaling [80], transcriptional control [81], and cancer cell survival [77], among many others [82,83]. Microproteins derived from noncanonical coding regions represent a substantial fraction of major histocompatibility complex (MHC) class I–associated tumor-specific antigens [84,85]. *De novo* appearance of short proto-genes coupled with rapid evolution may form a dynamic reservoir for genetic innovation and diversification [43]. Interestingly, lncRNAs appear to evolve rapidly from junk transcripts and, in turn, many lncRNAs appear to encode microproteins [86–88], consistent with the idea that such regulatory functions can readily emerge *de novo*. Fast evolving genes often have complex expression patterns compared to highly conserved genes and tend to be enriched for transcription-associated functions [89]. The rewiring of transcription factor function by microprotein partners may augment the inherent evolvability of transcriptional control by promoter binding site mutations [90]. As illustrated by the example of *NRS1*, transcription activation may be a particularly facile route for proto-genes to quickly acquire important regulatory functions that optimize fitness under specific conditions.

## Methods

### Yeast strains construction and culture

All *S. cerevisiae* strains were isogenic with the S288C (BY4741) auxotrophic background (S2 Table). Standard molecular genetics methods were employed for genomic integration of carboxyl-terminal tagging cassettes [91] and verified by PCR or sequencing. Standard media were used for yeast growth: rich (XY: 2% peptone, 1% yeast extract, 0.01% adenine, 0.02% tryptophan); synthetic complete (SC: 0.17% YNB, 0.2% amino acids, 0.5% ammonium sulfate); or nitrogen-limited (0.17% YNB, 0.4% proline, supplemented with histidine, leucine, methionine and uracil to complement auxotrophies as needed). Prototrophic *S. boulardii* strains were grown in SC or nitrogen-limited (0.17% YNB, 0.4% proline) without amino acid supplements. To delete *NRS1* in *S. boulardii*, the strain MYA-796 was transformed with pGZ110-Cas9-amd-SYM-nrs1_sgRNA and plated on acetamide-containing medium. Deletion alleles were confirmed by DNA sequencing of the coding region and the pGZ110 plasmid was counter-selected on fluoroacetamide medium [92]. Carbon sources were added to 2% w/v as indicated. Unless otherwise specified, cells were grown to saturation in SC + 2% glucose and diluted 1 in

5,000 into fresh medium for 16 to 18 hours growth prior to experiments. These conditions ensured homeostatic growth in log-phase at the time of experiment at a culture density of 2 to 5 $10^6$ cells/mL. For strains bearing constructs expressed under the inducible *GAL1* promoter, pre-growth was in SC + 2% raffinose. Cell size distributions were acquired using a Beckman Coulter Z2 counter (Beckman Coulter, Brea, CA, USA). Growth curves were acquired at 30˚C using a Tecan Sunrise shaker-reader (Tecan Group Ltd., Männedorf, Switzerland).

## SGA screen

A *cln3Δ bck2Δ whi5*::*GAL1-WHI5* strain was mated to an array of 5,280 *GAL1-ORF* fusion strains [41]. Following haploid selection, strains were scored for growth on selective medium containing 2% galactose. Three replicates of this screen were performed. For the third replicate, expression of the *GAL4* transcription factor was increased in case it was limiting for expression of *WHI5*, the ORFs, or *GAL* genes required for growth in the presence of galactose at the sole carbon source. For this purpose, *GAL4* was placed under the control of the *ADH1* promoter in a *cln3*::*LEU2 bck2*::*NAT^R whi5*::*Kan^R-pGAL1-WHI5 can1d mfa1*::*MFA1pr-spHIS5* + *GAL4p*::*HphRpADH1-GAL4* query strain. Screen hits were validated by transformation of the *GAL1-GST-ORF* construct directly into the *cln3Δ bck2Δ whi5*::*GAL1-WHI5* query strain.

## Chromatin immunoprecipitation

Cells were grown in XY with 2% raffinose, induced with 2% galactose for 6 hours to an $OD_{600}$ ≤0.5 and fixed with 1% formaldehyde. Whole-cell extracts from 50 mL of culture were prepared by glass bead lysis, sonicated to shear chromatin DNA into fragments, and incubated with the appropriate antibody coupled to magnetic beads (Dynabeads PanMouse IgG). Immunoprecipitated DNA was washed, de-crosslinked, purified, and analyzed by quantitative real-time PCR. Reactions with appropriate oligonucleotides were set-up with SYBR Green PCR Master Mix (Applied Biosystems, Waltham, Massachusetts, USA) and carried out on an ABI 7500 Fast Real-Time PCR System. Enrichment at the *CLN2* or *PCL1* locus was determined after normalization against values obtained from input samples using *SYP1* as the reference gene. [HA]Whi5 and Nrs1[13MYC] ChIP experiments were performed in biological duplicates. The bar height on Fig 6A and S4A Fig represent the mean over duplicates, and the error bars represent the standard error on the mean.

## Immunoprecipitation and immunoblot analysis

Protein extracts were prepared in lysis buffer (10 mM HEPES-KOH pH 7.9, 50 mM KCl, 1.5 mM $MgCl_2$, 1 mm EDTA, 0.5 mM DTT, 50 mM NaF, 50 mM sodium pyrophosphate, 1 mM $Na_3VO_4$ and Roche protease inhibitor cocktail; Roche, Basel, Switzerland) by glass bead lysis. On S3C Fig, the lysis buffer contained 200mM NaCl, 50 mM HEPES pH 7.5, 1 mM EDTA, 1 mM DTT, 1.5 mM $MgCl_2$, protease inhibitors, 2 mM NaF, and 2 mM Na-pyrophosphate. Due to low expression levels in the 60 to 150 nM range (Fig 2D and [22]), in some cases, we concentrated Nrs1[13MYC] by immunoprecipitation prior to immunoblot detection. Immunoprecipitations were carried out at 4˚C for 2 hours with indicated antibodies in soluble form (9E10 anti-MYC from EMD Millipore, Burlington, Massachusetts, USA, 05–419; M2 anti-FLAG from Sigma, St Louis, Missouri, USA, F1804) then bound to protein G resin (Pierce, Waltham, Massachusetts, USA, 20398) for 1 hour at 4˚ C, followed by either 1 or 3 washes with wash buffer (10 mM Tris-Cl pH7.9, 0.1% Triton X-100, 0.5 mM DTT, 0.2 mM EDTA, 10% glycerol, 150 mM NaCl; for Fig 5A, Wash buffer: 10 mM Tris-HCl pH7.9, 0.5 mM DTT, 0.2 mM EDTA, 150 mM NaCl), and resuspension in SDS sample buffer. In Fig 5A, the immunoprecipitations were performed as above except that cell pellets were resuspended in 200 mM NaCl, 50 mM

HEPES-KOH pH 7.9, 5mM EDTA, 1 mM DTT, MCE protease inhibitors, 2 mM NaF, 2 mM Na-pyrophosphate, 0.1% NP-40 prior to lysis and complexes captured directly onto 15 μL of M2 anti-FLAG resin (anti-FLAG M2 affinity gel Sigma, A2220). For S3C Fig, the complexes were captured directly onto 15 μL of anti-MYC resin (Pierce, 20168). Proteins were resolved by SDS-PAGE, transferred onto nitrocellulose membrane, and immunoblotted with anti-HA, anti-Swi4 (gift from Brenda Andrews), anti-Swi6 (gift from Kim Nasmyth), anti-GST, anti-MYC 9E10, or anti-FLAG M2 antibodies, followed by detection with the appropriate HRP-conjugated secondary antibody [12,93].

### *In vitro* binding assays

Recombinant GST-Nrs1 fusion protein was affinity purified with GSH-Sepharose 4B (Amersham Biosciences, Little Chalfont, Buckinghamshire, UK) in 50 mM HEPES-NaOH, pH 7.5, 150 mM NaCl, 5 mM EDTA, 5 mM NaF, 0.1% NP-40, 10% glycerol, supplemented with 1 mM PMSF. Recombinant SBF or MBF complex was purified with anti-FLAG resin (M2, Sigma) from insect cells co-infected with [FLAG]Swi4-Swi6 or [FLAG]Mbp1-Swi6 baculovirus constructs in buffer supplemented with complete protease inhibitor cocktail (Roche) and then eluted with excess FLAG peptide [12]. Binding reactions were incubated at 4˚C for 1 hour with rotation. Washed samples were resolved on SDS-PAGE gel followed by immunoblotting with anti-FLAG (M2, Sigma) and anti-Swi6. SBF-GST-Nrs1 and SBF-HA-Whi5 complexes were pre-formed in solution and immobilized on glutathione or anti-HA resin, respectively, incubated with GST-Nrs1 or [HA]Whi5, washed and analyzed as described above.

### Immunoprecipitation and mass spectrometry analysis

Cell pellets from untagged and Nrs1[13MYC] strains from 100 mL of culture at $OD_{600}$ = 1 were lysed in standard lysis buffer (50 mM Tris-HCl pH8.0, 150 mM KCl, 100 mM NaF, 10% glycerol, 0.1% tween-20, 1 mM tungstate, 1 mM DTT, 10 mM AEBSF, 10 mM pepstatin A, 10 mM E-64) [56] supplemented with protease inhibitors using a $N_2$(l) freezer mill. Lysates (0.5 mL) were incubated for 1 hour with 5 μL of anti-MYC antibody (Gentex, Zeeland, MI, USA) followed by 1-hour incubation with an additional 50 μL of GammaBind plus Sepharose beads (GE Healthcare, Chicago, IL, USA) to capture protein complexes. After multiple washes, samples were separated on Bio-Rad (Hercules, CA, USA) precast gels, and the entire gel lane was cut for each sample (CAPCA core facility, https://capca.iric.ca; see Supporting information Methods). Gel samples were destained, alkylated, and digested with trypsin for 8 hours at 37˚C and peptides extracted in 90% ACN. Peptides were separated on a home-made C18 column connected to Q-Exactive HF Biopharma with a 56-minute gradient of 0% to 30% acetonitrile in 0.2% formic acid. Each full MS spectrum of extracted peptides was acquired at a resolution of 120,000, followed by acquisition of 15 tandem MS (MS–MS) spectra on the most abundant multiply charged precursor ions by collision-induced dissociation (HCD). Data were processed using PEAKS X (Bioinformatics Solutions, Waterloo, Ontario, Canada) and the UniProt yeast database. Variable selected posttranslational modifications were carbamidomethyl (C), oxidation (M), deamidation (NQ), acetyl (N-ter), and phosphorylation (STY). Control untagged and Nrs1[13MYC] data were analyzed with Scaffold 4.8.9 at an FDR of 1% for at least 2 peptides with a likelihood of at least 99%.

### RNA-seq analysis

Wild-type and *GAL1-NRS1* strains were inoculated for overnight growth in SC + 2% raffinose at 30˚C, then diluted in SC + 2% raffinose to an OD600 of 0.07–0.1, incubated at 30˚C until an OD600 of 0.15–0.2 was reached, at which point a sample was collected for RNA extraction

prior to galactose induction. For *NRS1* induction, the cultures were either concentrated by centrifugation, resuspended in 20 mL SC+2% raffinose and added to 200 mL SC+2% galactose (replicate 2,3) or grown to an OD600 of 0.15 to 0.2 in 200 mL SC + 2% raffinose followed direct addition of 2% galactose to the culture (replicate 1). Time point 0 was collected immediately, followed by time points up to 6 hours. Total RNA was prepared as described in [94]. Briefly, cells were pelleted at 3500 rpm, disrupted using glass beads, phenol/chloroform extracted, and the aqueous phase containing nucleic acids was ethanol-precipitated. Contaminating genomic DNA was removed by DNase treatment (Qiagen, Hilden, Germany). Total RNA was quantified using Qubit (Thermo Scientific, Waltham, MA, USA) and RNA quality assessed with a 2100 Bioanalyzer (Agilent Technologies, Santa Clara, CA, USA). Transcriptome libraries were generated using the Kapa RNA HyperPrep (Roche) using a Poly-A selection (Thermo Scientific). Sequencing was performed on the Illumina NextSeq 500 system, aiming for 10 million single-end reads per sample, 40 million for raffinose, and final (6 hours) induced time points. RNA-seq experiments were performed in biological triplicate.

Reads were aligned to all Ensembl yeast transcripts using Bowtie 2.2.5 [95] with default parameters. Read counts were tabulated for each gene, only considering alignments with an edit distance no greater than 5. Genes with at least 500 reads in at least 1 sample were included in the analysis. Gene expression levels for each sample, expressed as log2 read counts, were normalized to the upper quartile gene expression level for the same sample, as recommended in Bullard and colleagues [96], and represented as heatmaps Fig 6C.

For each of the 3 *GAL1-NRS1* biological replicate samples, the expression after 6 hours galactose induction was normalized separately to the expression in raffinose (prior to induction) and to the wild-type sample from the same replicate after 6 hours induction. The former controls for the sample-to-sample variation, while the latter controls for the effect of the switch to galactose-containing media. To ensure to the best possible extent that observed expression differences are due to *NRS1* expression and not to one of the 2 abovementioned sources of variation, the direction and magnitude of the differential expression of each gene had to be reproduced relative to both controls. We thus kept as the final score for each replicate the smallest absolute log2-fold change produced by either of the 2 comparisons and set the score to zero if the signs of the 2 disagreed. The values for each of the 3 replicates were averaged to yield the final gene score (S6 Table). This procedure was designed to control for the effects of galactose medium (i.e., normalizing by 6 hours wild-type samples) and for intersample expression variability (i.e., normalization to expression levels prior to induction of each culture and averaging over replicates). Hence, high scores for *GAL1-NRS1* (positive or negative) could be obtained only for genes up-/down-regulated both in galactose compared to raffinose and compared to wild-type cells in galactose. Wild-type 6 hours scores (S6 Table) were obtained in the same fashion except that gene expression in wild-type cells after 6 hours in galactose were normalized separately to wild-type expression in raffinose and to *GAL1-NRS1* after 6-hour induction. This control analysis is symmetrical and thus equivalent to the analysis performed on *GAL-NRS1* samples.

## High-content imaging

High-content images were acquired on an OPERA high-throughput confocal microscope (PerkinElmer, Waltham, MA, USA) equipped with a 60× water objective. Pixel resolution was 220 nm at 2*2 binning. 200 μL of culture were directly transferred to a Greiner Screenstar glass-bottom 96-well imaging plate and imaged within 30 to 45 minutes. GFP was excited with a 488 nm laser (800 ms exposure for endogenously tagged proteins and 320 ms for overexpressed proteins) and detected using a 520/35 nm band-pass filter. For *WHI5-GFP* and

*WHI5-NRS1-GFP* strains, pre-Start G1 phase cells were identified by nuclear localization of the GFP signal, as computed using a custom image-analysis script written in MATLAB (Math-Works, Natick, MA, USA). The script masks individual cells using threshold-based detection of the autofluorescence background and determines the fraction of individual cells that display a pre-Start nuclear localization of the GFP signal.

### sN&B and RICS fluorescence fluctuation microscopy

Live cells in log-phase were imaged on an Alba sN&B system (ISS, Champaign Illinois, USA) comprised of an inverted confocal Nikon Eclipse microscope equipped with a 100× water objective, a Fianium Whitelase continuous white laser with 488 nm emission filters and single photon APD detectors, as described previously [22]. In brief, 1.5 mL of cell culture was pelleted for 2 minutes at 3,000 rpm in a microfuge, resuspended in approximately 100 μL of culture supernatant and 3 μL deposited on a preset 65 μL drop of SC + 2% glucose + 2% agar gel medium on a circular glass coverslip (#1, VWR) that was encircled by an adhesive silicon ring. After 4 minutes drying time, a ConcanavalinA-coated (Sigma, 2 mg/mL) coverslip was gently pressed on top of the agarose to seal the pad against the adhesive silicon ring. Sealed pads were clamped in an AttoFluor chamber (Molecular Probes, Eugene, OR, USA) and immediately imaged for no more than 1.5 hours. Unless otherwise specified, the culture growth medium was reused to make the imaging pads in order to prevent inadvertent nutrient up- or down-shifts. For this purpose, 1 mL of cell culture was pelleted at 15,000 rpm for 1 minute, and 500 μL of the supernatant was mixed with 10 mg of agarose and warmed for 1 to 2 minutes at 98˚C to melt the agar, followed by application to a coverslip. For sN&B experiments in the presence of rapamycin, growth medium containing 100 nM rapamycin was also used to prepare the pad. To mitigate possible degradation of rapamycin during the pad preparation, an additional 100 nM of rapamycin was added to the pre-warmed agar mix just before making the pad. The final rapamycin concentration was therefore 200 nM, i.e., about 200 ng/mL, similar to rapamycin treatment time courses in Figs 5B and S1C, and the fixed-duration treatment in Fig 5A. Cells were pretreated 1 hour before imaging and imaging lasted about 1.5 hours, yielding a treatment duration between 1 hour and 2.5 hours across the various fields of view (FOVs). There were no significant differences in Nrs1-GFP signals across different FOVs.

sN&B imaging was performed using 20 raster scans of the same 30 μm–wide FOVs of 256 pixels (pixel size 117 nm), using an excitation power of 1 to 2 μW at 488 nm wavelength and a 64 μs pixel dwell time. sN&B images shown in the figures are projections of the 20 raster scans. Protein concentrations were extracted from sN&B data using custom analysis software [22]. RICS imaging was performed in a similar fashion to sN&B imaging but different parameters were used to improve correlation curves (pixel size of 48.8 nm, 50 frames, 20 μs pixel dwell time) as described previously [56]. RICS vertical correlations for individual FOVs were fitted to single mode–free diffusion models using the SimFCS analysis software. FOVs of poor-quality fit were discarded from diffusion coefficients plots.

### Bioinformatics

The sequences of orthologs of *YLR053c* were obtained from the Fungal Orthogroups website. For *Saccharomyces paradoxus* and *Saccharomyces mikatae*, where no ORF was predicted at the expected locus, the *YLR053c* sequence was searched against whole genome assemblies using TBLASTN 2.2.27 to predict orthologs. The synteny of the regions in *K. waltii* and *S. cerevisae* was confirmed based on Orthogroups orthology assignments of the upstream (47.17997 and *IES3*) and downstream (47.18006 and *OSW2*) genes and the fact that each of these assignments

were supported by highly significant Blast e-values. Alignments were generated using ClustalO 1.2.0 and displayed using the MView online tool.

## Mathematical model of Start

Whisker and box plots for model predictions of the critical size at Start were obtained using the mathematical model and the data processing procedure described previously [22]. For each plot, 50 individual cells bearing variable Whi5 concentrations randomly picked in the 100 to 140 nM (for the *WHI5-GFP* strain) and 65 to 105 nM (for the *WHI5-NRS1-GFP* strain) ranges were simulated and plots generated in Excel using standard statistics of the distributions of critical size values for each strain.

## Supporting information

**S1 Fig. Additional characterization of Nrs1. (A)** Full Ylr053c/Nrs1 protein sequence is conserved across the *Saccharomyces sensu stricto* group of species. Sequence alignment showing the Ylr053c/Nrs1 protein sequence in *S. cerevisiae* (top), aligned with sequences of orthologs in *S. bayanus* (c672-g32.1), *S. castellii* (656.13d), *S. paradoxus*, and *S. mikatae* from top to bottom. Ylr053c/Nrs1 orthologs were not predicted in *S. paradoxus* or *S. mikatae* because of sharp length cutoffs in ORF prediction algorithms (the ORFs would span only 108 and 75 residues in *S. paradoxus* and *S. mikatae*, respectively). The lack of an obvious TATA box could also explain why no protein was predicted in *S. paradoxus*. Neighboring upstream and downstream genes both show high similarity to *YLR053c* neighbors in *S. cerevisiae*. The *YLR053c/ NRS1* sequence also aligns in *S. kudriavzevii* (not shown). **(B)** Kinetics of Nrs1 expression upon nitrogen starvation. sN&B images of untagged WT and *NRS1-GFP* log-phase cells (OD = 0.4 to 0.7), following 22 hours growth in YNB Pro medium from 1/5,000 dilution (left) and 7 hours growth from 1/100 dilution (right) from saturated precultures. Arrows indicate representative Nrs1 signal beyond autofluorescence at 22 hours. **(C)** Prolonged exposure to rapamycin results in accumulation of a faster migrating form of Nrs1. Rapamycin was added to log-phase cultures of *NRS1^13MYC^* cells, aliquots were removed at indicated time intervals, and immunoprecipitates were analyzed by anti-MYC immunoblot. A raw image of the original immunoblot is provided in the S1 Raw Images. **(D)** Nuclear localization of Nrs1 upon rapamycin treatment is not a consequence of the particular GFPmut3 fluorophore. Confocal microscopy image of *NRS1^WT-GFP^* cells grown in SC + 2% glucose medium, either untreated or treated with 200 ng/mL rapamycin for 2 hours. **(E)** sN&B images of untagged WT cells and *NRS1-GFP* cells grown to log-phase in SC + 2% glucose and plated on SC + 2% glucose agar pads containing either 0.5 M NaCl or 1 mM $H_2O_2$ and imaged over a 2-hour time course. Images were acquired after approximately 1-hour treatment, but Nrs1 expression was not observed at any time point for any of the treatments. **(F)** Nrs1 is not induced by DNA damage. *NRS1^13MYC^* cells grown in rich medium were exposed 0.1% MMS for 1 hour prior to immunoprecipitation and *Nrs1^13MYC^* was detected with anti-MYC 9E10 antibody. Nrs1 expression from cells exposed to 200 ng/mL rapamycin and processed in parallel served as a positive control. A raw image of the original immunoblot is provided in the S1 Raw Images. *NRS1*, Nitrogen-Responsive Start regulator 1; sN&B, scanning Number and Brightness; WT, wild-type. (PDF)

**S2 Fig. Additional characterization of *nrs1Δ* strains. (A)** Deletion of *NRS1* does not affect growth. Optical density (vertical axis) of WT (black) and *nrs1Δ* (blue) strains grown in SC + 2% glucose (solid lines) or nitrogen-limited (YNB+Pro, dashed lines) medium as a function of time (horizontal axis). **(B–D)** Deletion of *NRS1* does not affect cell size. Cell size

distributions of WT (black) and *nrs1Δ* (blue) strains grown in SC + 2% glucose (**B**, solid lines), nitrogen-limited (**B**, YNB+Pro, dashed lines), SC + 2% galactose (**C**, solid lines), SC + 2% raffinose (**C**, dotted lines), YNB + 0.4% proline + 2% galactose (**C**, YNB pro gal, dashed lines), SC + 2% glucose (**D**, solid lines), SC + 4% glucose (**D**, dotted lines) and SC + 0.1% glucose (**D**, dashed lines). **(E)** Deletion of *NRS1* does not affect competitive fitness in SC + 2% glucose and nitrogen limited (YNB+Pro) medium during growth to stationary phase. Bar charts representation of the composition of 2 mixes of competing strains (Mix1: WT transformed with mCherry plasmid (red) and *nrs1Δ* transformed with Venus plasmid (green); Mix2: WT with Venus plasmid (green), and *nrs1Δ* with mCherry plasmid (red)) as a function of time from inoculation. The percentage of each strain within the mixes shown is derived from 3 replicate cultures from the same original mixes (see S1 Text Methods). Error bars show the standard error on the mean. All numerical values underlying this figure may be found in S2 Data. *NRS1*, Nitrogen-Responsive Start regulator 1; WT, wild-type.
(PDF)

**S3 Fig. Additional data demonstrating the physical interaction between Nrs1 and SBF. (A)** Example Swi4 and Swi6 peptide spectra detected in Nrs1 immunoprecipitates. The first of the 5 peptides identified for Swi4 (ITSPSSYNKTPR) and Swi6 (SGLRPVDFGAGTSK) are shown on top and bottom, respectively. Data were processed with Scaffold software. **(B)** Replicate experiment for interactions detection with endogenous level of tagged proteins (Fig 5A). Swi4[3FLAG] or Swi6[3FLAG] complexes were immunoprecipitated from the indicated strains grown in the presence of 200 nM rapamycin for 3 hours and interacting proteins assessed by immunoblot with the indicated antibodies. Co-immunoprecipitation of Whi5[13MYC] with Swi4[3FLAG] and Swi6[3FLAG] served as a positive control. Mr markers (M) are indicated for each blot. Pgk1 served as a loading control. Asterisk indicates IgG heavy and light chains. **(C)** Detection of endogenous untagged Swi4 and Swi6 in Nrs1[13MYC] immunoprecipitates. Nrs1[13MYC] or Whi5[13MYC] complexes were immunoprecipitated from cultures of the indicated strains that were either untreated or treated with 200 nM rapamycin for 3 hours. Interacting proteins assessed by immunoblot with the indicated antibodies. Co-immunoprecipitation of Swi4 and Swi6 with Whi5[13MYC] served as a positive control. Double asterisk indicates Whi5[13MYC] degradation product that migrated at a similar size as Nrs1[13MYC]. Mr markers (M) are indicated for each blot. *NRS1*, Nitrogen-Responsive Start regulator 1; SBF, SCB-binding factor. (PDF)

**S4 Fig. Additional data demonstrating lack of effect of Nrs1 on Whi5. (A)** *NRS1* overexpression does not inhibit Whi5 association with G1/S promoter DNA. WT or *WHI5[HA]* strains carrying empty vector or a *GAL1-NRS1* plasmid were grown in SC + 2% raffinose medium and induced with 2% galactose for 6 hours prior to crosslinking. Anti-HA ChIPs were assessed for the presence of *CLN2* and *PCL1* promoter DNA by quantitative RT-PCR. Bars indicate the mean fold-enrichment across 2 replicates, and error bars show the standard error on the mean. **(B)** *NRS1* overexpression does not affect Whi5 protein levels. Whi5-GFP absolute concentration in single WT (blue dots) and *GAL1-NRS1* (orange dots) cells first grown in SC + 2% raffinose then induced with 2% galactose for 6 hours prior to sN&B microscopy. Nuclear Whi5-GFP levels in pre-Start cells and cell-averaged levels in post-Start cells where Whi5 has been exported from the nucleus are shown. All numerical values underlying panels A and B may be found in S4 Data. **(C)** *NRS1* overexpression does not inhibit Whi5 association with SBF. The indicated Whi5[HA] immunoprecipitates from strains induced with galactose for 6 hours were probed for Whi5[HA], Swi4, or Swi6 by immunoblot. **(D)** Nrs1 does not compete with Whi5 for binding to SBF *in vitro*. The indicated amounts of recombinant [HA]Whi5 or [GST]Nrs1 was titrated into preformed [FLAG]Swi4-Swi6-[GST]Nrs1 or [FLAG]Swi4-Swi6-[HA]Whi5

complexes immobilized on anti-FLAG resin, respectively. Bound proteins were resolved by SDS-PAGE then immunoblotted (top) or stained with Coomassie Brilliant Blue (bottom). Note that added soluble $^{GST}$Nrs1 or $^{HA}$Whi5 saturated the respective SBF-$^{HA}$Whi5 and SBF-$^{GST}$Nrs1 complexes at the lowest input concentrations. Raw image of the original immunoblots used to made panels C and D are provided in S1 Raw Images. ChIP, chromatin immunoprecipitation; *NRS1*, Nitrogen-Responsive Start regulator 1; SBF, SCB-binding factor; sN&B, scanning Number and Brightness; WT, wild-type.
(PDF)

**S5 Fig. Additional data relevant to Nrs1-mediated transactivation and *GAL1-NRS1* genetic interactions. (A)** Control growth curves for transactivation assays. Reporter strains transformed with plasmids expressing either *GAL4$^{DBD}$* alone, *GAL4$^{DBD}$-NRS1*, *GAL4$^{DBD}$-UBE2G2*, or *GAL4$^{DBD}$-NRS1$^{Cter}$* were grown in SD-Trp medium at 30˚C. **(B)** *NRS1* overexpression does not rescue a *cln1Δcln2Δcln3Δ* G1 phase arrest. Left: Cultures of *cln1Δcln2Δcln3Δ MET-CLN2* and *cln1Δcln2Δcln3Δ MET-CLN2* + *<pGAL1-NRS1>* strains grown to log-phase in SC-Met +2% raffinose, then reinoculated in either SC-Met+2% raffinose, SC-Met+2% galactose, SC+Met+2% raffinose or SC+Met+2% galactose for the indicated periods of time before determination of cell size distributions on a Beckman Z2 Coulter counter. Right: bar charts showing the average number of cell divisions for *cln1Δcln2Δcln3Δ MET25-CLN2* and *cln1Δcln2Δcln3Δ MET25-CLN2* + *<pGAL1-NRS1>* strains during the 18 hour interval between the 6 hours and 24 hours time points in SC+Met+2%galactose. Bar heights represent the average of 4 different clones (*N* = 4); error bars represent the standard deviation. **(C)** Room temperature growth controls for genetic interactions of *NRS1* with *SWI4* and *MBP1*. Serial 5-fold dilutions of WT *NRS1* and *nrs1::GAL1-NRS1* strains in WT (rows 1, 2, 10), *swi4-ts* (row 3), *mbp1Δ* (row 4), and *mbp1Δswi4-ts* (rows 5–9) backgrounds were spotted onto SC + 2% glucose, SC + 2% raffinose, and SC + 2% galactose medium and grown for 5 days at 23˚C. C1-4 are 4 clones of *mbp1Δ swi4-ts GAL1-NRS1*. **(D)** Images of the same serial 5-fold dilutions of *NRS1* and *GAL1-NRS1* strains in WT, *swi4-ts*, *mbp1Δ*, and *mbp1Δ swi4-ts* backgrounds as in Fig 7B, spotted onto SC + 2% glucose, SC + 2% raffinose and SC + 2% galactose, but grown for an additional 2 days (i.e., 7 days total growth time at 30˚C). C1 to C4 are 4 clones of *mbp1Δ swi4-ts GAL1-NRS1*. All numerical values underlying panels A and B may be found in S6 Data. *NRS1*, Nitrogen-Responsive Start regulator 1; WT, wild-type.
(PDF)

**S6 Fig. Additional characterization and controls for Nrs1 and Whi5 fusion proteins. (A)** The Whi5-Nrs1-GFP chimeric protein is produced *in vivo* and migrates at the expected size. *WHI5-GFP*, *NRS1-GFP*, and *WHI5-NRS1-GFP* strains were grown in nitrogen-limited (YNB +Pro) medium and extracts immunobloted with anti-GFP antibody. A raw image of the original immunoblot is provided in S1 Raw Images. **(B)** A carboxyl-terminal fusion of Nrs1 to Whi5 does not affect cell cycle distribution. High-content images of *WHI5-GFP* and *WHI5-NRS1-GFP* cells grown in SC + 2% glucose were acquired on an OPERA high-throughput confocal microscope (PerkinElmer) equipped with a 60× water objective. The same intensity scale was used for both panels. Scale bar is 10 μm. The fraction of pre-Start (G1) cells was obtained using a custom MATLAB script (see Methods). **(C)** Fusion of a GFP tag at the Whi5 carboxyl terminus does not affect cell size. Cell size distributions of untagged WT and *WHI5-GFP* cells grown in SC + 2% glucose were determined on a Beckman Z2 Coulter counter. **(D)** Growth curves of WT, *whi5Δ*, and *WHI5-NRS1-GFP* strains in SC + 2% glucose medium at 30˚C. **(E)** *WHI5* dosage has only minor effects on cell size. Cell size distributions of WT and *WHI5/whi5* heterozygous diploid strains grown in SC + 2% glucose. Dotted, dashed, and solid blue lines represent 3 different *WHI5/whi5* clones. **(F)** Predicted effects of *WHI5*

dosage on cell size in a mathematical model of Start. Box and whisker plots show distribution of critical cell sizes predicted by the Start model published in [22] for simulated average Whi5 concentrations of 120 nM (corresponding to *WHI5-GFP* cells, left boxplot) and 85 nM (corresponding to *WHI5-NRS1-GFP* cells, right boxplot). All numerical values underlying panels C–F may be found in S7 Data. *NRS1*, Nitrogen-Responsive Start regulator 1; WT, wild-type; YNB +Pro, YNB + 0.4% proline + 2% glucose.
(PDF)

**S7 Fig. Nrs1 function at G1/S requires the poorly conserved N-terminal region. (A)** Cell size distributions of WT and *WHI5-NRS1^{Cter}-GFP* strains grown in SC+2% glucose determined on a Beckman Z2 Coulter counter. **(B)** Genotype of 10 tetrads from a *cln3Δ whi5::WHI5-NRS1^{Cter}-GFP* X *bck2Δ* cross. For each tetrad, spore clone growth was assessed on SD-Leu (indicates *cln3::LEU2*), SC+NAT (indicates *bck2::NAT^R*), and SD-HIS (indicates *whi5::WHI5-NRS1^{Cter}-GFP-HIS3*). Blue boxes indicate viable *cln3Δ* or *bck2Δ* spore clones. No viable *cln3Δ bck2Δ* double mutant clones were recovered. All numerical values underlying panel A may be found in S7 Data. *NRS1*, Nitrogen-Responsive Start regulator 1; WT, wild-type.
(PDF)

**S1 Table. Candidate dosage suppressors of *cln3Δ bck2Δ* lethality.**
(XLSX)

**S2 Table. Yeast strains and plasmids used in this study.**
(DOCX)

**S3 Table. Proteins detected specifically in Nrs1 immunoprecipitates by mass spectrometry.** *NRS1*, Nitrogen-Responsive Start regulator 1.
(XLSX)

**S4 Table. List of peptides identified in Nrs1 and control immunoprecipitates by mass spectrometry.** *NRS1*, Nitrogen-Responsive Start regulator 1.
(XLSX)

**S5 Table. List of proteins identified in Nrs1 and control immunoprecipitates by mass spectrometry.** *NRS1*, Nitrogen-Responsive Start regulator 1.
(XLSX)

**S6 Table. Calculated gene scores for RNA-seq experiments performed with WT and *GAL1-NRS1* strains.** For up-regulated (score >0) and strongly up-regulated (score > 0.5) genes, Swi4/Mbp1-binding scores were imported from Ferrezuelo and colleagues [10], and SBF/MBF targets were identified and counted. These data were used to perform the hypergeometric enrichment tests. MBF, MCB-binding factor; *NRS1*, Nitrogen-Responsive Start regulator 1; RNA-seq, RNA sequencing; SBF, SCB-binding factor; WT, wild-type.
(XLSX)

**S1 Raw Images. Contains raw images of uncropped gels (western blots, Coomassie, Ponceau) used in the manuscript.**
(PDF)

**S1 Data. Raw numerical data used to generate Fig 2D and 2E.**
(XLSX)

**S2 Data. Raw numerical data used to generate Figs 3 and S2.**
(XLSX)

**S3 Data. Raw numerical data used to generate Fig 4A and 4B.**
(XLSX)

**S4 Data. Raw numerical data used to generate S4A and S4B Fig.**
(XLSX)

**S5 Data. Raw numerical data used to generate Fig 6A and 6B.**
(XLSX)

**S6 Data. Raw numerical data used to generate Figs 7A and S5A and S5B.**
(XLSX)

**S7 Data. Raw numerical data used to generate Figs 8A–8C and S6C–S6F and S7A.**
(XLSX)

**S1 Text. Supporting information.**
(PDF)

## Acknowledgments

We thank Marc Angeli for technical assistance with SGA screens, Brenda Andrews and Kim Nasmyth for providing antibody reagents and strains, and Elizabeth Bilsland for yeast fluorescent protein vectors. We also thank Jennifer Huber of the IRIC Genomics Platform for assistance with RNA-seq experiments.

## Author Contributions

**Conceptualization:** Sylvain Tollis, Jaspal Singh, Mike Tyers.

**Data curation:** Sylvain Tollis, Jasmin Coulombe-Huntington, Eric Bonneil.

**Formal analysis:** Sylvain Tollis, Jasmin Coulombe-Huntington, Eric Bonneil.

**Funding acquisition:** Sylvain Tollis, Pierre Thibault, Mike Tyers.

**Investigation:** Sylvain Tollis, Jaspal Singh, Roger Palou, Yogitha Thattikota, Ghada Ghazal, Xiaojing Tang, Susan Moore, Deborah Blake.

**Methodology:** Sylvain Tollis, Jaspal Singh, Xiaojing Tang, Susan Moore, Catherine A. Royer, Mike Tyers.

**Project administration:** Sylvain Tollis, Mike Tyers.

**Resources:** Sylvain Tollis, Jaspal Singh, Roger Palou, Yogitha Thattikota, Ghada Ghazal, Xiaojing Tang, Susan Moore, Deborah Blake.

**Software:** Sylvain Tollis, Jasmin Coulombe-Huntington.

**Supervision:** Sylvain Tollis, Mike Tyers.

**Validation:** Sylvain Tollis, Jaspal Singh, Roger Palou, Yogitha Thattikota, Ghada Ghazal, Xiaojing Tang, Susan Moore, Deborah Blake.

**Visualization:** Sylvain Tollis, Jaspal Singh, Roger Palou, Ghada Ghazal, Jasmin Coulombe-Huntington, Eric Bonneil.

**Writing – original draft:** Sylvain Tollis, Jaspal Singh, Deborah Blake, Eric Bonneil.

**Writing – review & editing:** Sylvain Tollis, Roger Palou, Yogitha Thattikota, Ghada Ghazal, Jasmin Coulombe-Huntington, Catherine A. Royer, Pierre Thibault, Mike Tyers.

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
