## [Editor Report · Decision Letter 0]

6 Jan 2021

Dear Dr Tollis, 

Thank you for submitting your manuscript entitled "A nitrogen source-regulated microprotein confers an alternative mechanism of G1/S transcriptional activation in budding yeast" for consideration as a Research Article by PLOS Biology. Please accept my apologises for the delay in getting back to you due to the closure of the editorial office over the holiday period. 

Your manuscript has now been evaluated by the PLOS Biology editorial staff as well as by an academic editor with relevant expertise and I am writing to let you know that we would like to send your submission out for external peer review.

Please re-submit your manuscript within two working days, i.e. by Jan 08 2021 11:59PM.

Kind regards,

Richard Hodge, PhD

Associate Editor

PLOS Biology

---

## [Decision Letter · Decision Letter 1]

29 Jan 2021

Dear Dr Tollis,

Thank you very much for submitting your manuscript "A nitrogen source-regulated microprotein confers an alternative mechanism of G1/S transcriptional activation in budding yeast" for consideration as a Research Article at PLOS Biology. Your manuscript has been evaluated by the PLOS Biology editors, an Academic Editor with relevant expertise, and by several independent reviewers.

The reviews are attached below. You will see that the reviewers find your conclusions novel and interesting, but they also raise overlapping concerns about the reliance on NRS1 overexpression phenotypes presented in the manuscript and the media used during the experiments.

In light of the reviews, we will not be able to accept the current version of the manuscript, but we would welcome re-submission of a much-revised version that takes into account the reviewers' comments. We cannot make any decision about publication until we have seen the revised manuscript and your response to the reviewers' comments. Your revised manuscript is also likely to be sent for further evaluation by the reviewers…

IMPORTANT: We ask that you please fully address the experimental revisions proposed by Reviewer #1 and #3 regarding the lack of a loss-of-function phenotype and the comments provided by Reviewer #3 on the medium used in the experiments.

We expect to receive your revised manuscript within 3 months. 

**IMPORTANT - SUBMITTING YOUR REVISION**

*Re-submission Checklist*

*Published Peer Review*

*PLOS Data Policy*

*Blot and Gel Data Policy*

Sincerely,

Richard

Richard Hodge, PhD

Associate Editor

PLOS Biology

REVIEWS:

Reviewer's Responses to Questions

PLOS authors have the option to publish the peer review history of their article (what does this mean?). If published, this will include your full peer review and any attached files.

Reviewer #1: No

Reviewer #2: No

Reviewer #3: No

Reviewer #4: No

Reviewer #1: These authors are interested in identifying over-expressed genes that enable cells to Start the yeast cell cycle in the absence of two positive regulators (Cln3 and Bck2) and the presence of an excess of the negative regulator (Whi5). They screened a library that ectopically expresses genes at a very high level for rescue of this strain, and found 12 candidates. Two were expected (Cln1 and Cln3) and the others were either unknown functions or have no direct connection to Start at the moment. Among these is NRS1, which is an uncharacterized gene with promising characteristics in that it directly associates with the G1/S transcription complex SBF (Swi4/Swi6) and can function as a transcription activator when tethered to DNA. Nrs1 is only detected in nitrogen limitation or TOR inhibition (rapamycin treatment), hence the name Nitrogen Responsive Start regulator. Chromatin IP shows only a two-fold increase in Nrs1 on SBF target promoters that is rapamycin-dependent, so maybe it binds these promoters, but there is no transcription data to support that it activates these genes. They disprove the hypothesis that Nrs1 competes with the negative regulator (Whi5.) Most of the data supporting a role for Nrs1 in promoting start relies on ectopic over-expression. The nrs1 deletion has no effect on growth rate or cell size in any medium including nitrogen limitation. There are no tested conditions in which nrs1- delays Start or affects fitness. The over-expressed GAL:NRS1 is smaller than wild type, suggesting an acceleration of the G1 to S transition, and this size shift requires Swi4 and Swi6. GAL:NRS1 is also smaller than cln3 mutants. The GAL:NRS1 rescue of swi4ts mbp1 (no activators present) is very slight but the rescue of SWI6SA4 GAL:WHI512A (which leaves the repressive Whi5 on the SBF complex) is quite convincing. 

Overall, Nrs1 is an interesting small (108aa) protein that directly associates with key Start-regulators for which there is no tangible role at this time. Their model is that Nrs1 bypasses the normal Start controls under nitrogen limitation—but why would cells do that? The work is of high quality and a lot of effort has been made, but their conclusions rely heavily on overproduction phenotypes. Two resources that would help firm these conclusions would be flow cytometry to show that the G1 to S transition is affected by Nrs1, and transcript measurements demonstrating its activatory role at the anticipated target genes. 

Other comments to improve the manuscript:

1. The chromatin IPs show two-fold effects. There are no error bars and no information about number of replicates.

2. Rescue of swi4ts mbp1 by GAL:NRS1 is almost invisible in Figure 5B and can't be seen in Fig S4B. Can you get a better rescue at a lower temperature or with just swi4ts at a higher temperature? What do C2, C7 etc refer to?

3. The size differences in Fig 6C are very small and reflect a broadening of size distribution more than a significant difference.

4. SGD reports that over-expression of Nrs1 slows vegetative growth—is that a strain difference?

5. You didn't get BCK2 or CLN2 in your initial high copy screen. Do they rescue?

Reviewer #2: The paper by Tollis et al reports a new transcriptional activator (Nrs1), which activates G1/S transcription in budding yeast under nitrogen limitation. The manuscript is a revised version. This reviewer did not see the original manuscript. Overall, the discovery of a new regulator of G1/S transcription, especially one that appears to be sensitive to nutrient limitations, would be a significant finding, in this heavily studied field.

The authors did an over-expression screen, and found that Nrs1 in high dosage (its expression induced by galactose in this case) enabled cells that lacked other regulators to go through the G1/S transition. It also strongly reduced cell size, consistent with its role as an activator of Start. A variety of follow-up work is supportive of such a role. Overall, the work is well done and I have no issues with technical aspects.

My main concern is that of significance and physiological relevance. All the results deal with over-expression of Nrs1. Apparently, deletion of Nrs1 has no effect under any nutrient conditions (including when it is normally expressed, in poor nitrogen sources). What we are left with is a phenotype evident only when a poorly conserved short protein is over-expressed. Furthermore, in the vast majority of the assays in the figures, the experiments were done in media in which Nrs1 is not even normally present (because Nrs1 is normally around only under nutrient limitation). Can one then confidently conclude that just because ectopic expression of Nrs1 can do things at G1/S, then Nrs1 normally does the same things in wild type cells? 

Reviewer #3: In this interesting manuscript Tollis et al. describe Nrs1, a new budding yeast cell cycle regulator that promotes passage through Start in G1 under nitrogen limiting conditions. NRS1 (Nitrogen-Responsive Start Regulator 1) encodes a small nuclear protein that it is only expressed when TORC1 is downregulated, either by Rapamycin treatment or by growing the yeasts cells in nitrogen-poor media. Nrs1 interacts genetically and physically with SWI4 and SWI6, which encode components of the Start transcription factor SBF. Consistent with all of these findings, overexpression of NRS1 causes a small cell size phenotype. 

Most experiments in this manuscript are well executed and the data is very clear. However, there are some experimental details and comments that need to be addressed before accepting this manuscript for publication in PLOS Biology.

Experimental detail:

Table S2 and page 13. Are the strains used in this study prototrophic? This is an important point because the supplements (histidine, leucine, methionine, uracil,…) that have to be added to the minimal medium (YNB) are nitrogen sources that activate TORC1 (Stracka 2014, JBC 289: 25010-25010).

As described on page 13 (last paragraph), the nitrogen-limited media used in this manuscript contains supplements: "... nitrogen-limited (0.17% YNB, 0.4% proline, supplemented with histidine, leucine, methionine and uracil to complement auxotrophies as needed)". This point has to be clarified and key experiments should be repeated using prototrophic strains. 

Major points: 

1. If Nrs1 is a positive regulator of START in nitrogen-poor media, then nrs1∆ cells should show a delay in G1. This could be tested using prototrophic strains in YNB-proline without supplements and in other nitrogen-poor media, such as YNB-phenylalanine or YNB-isoleucine.

2. Does Nrs1 inhibit the interaction of Whi5 to the SBF promoters in nitrogen-poor media without supplements?

3. Page 10, last paragraph. The authors construct a strain expressing the fusion protein Whi5:Nrs1:GFP from the endogenous WHI5 promoter. These cells showed a reduced cell size similar to whi5∆ cells. One possible explanation for this phenotype and other phenotypes described in the manuscript could be that Whi5:Nrs1:GFP is a non-functional protein and behaves like the WHI5 deletion.

Other comments:

1. In most western blots (Figs. 2B, 4A, S1E), the authors immunoprecipitate the proteins before doing the immunoblot. Is this necessary because Nrs1, Whi5, Swi4 and Swi6 protein levels of are very low? This should be mentioned in the text.

2. Genetic interactions: 

* Are the swi4∆ nrs1∆, swi4ts nrs1∆ and swi6∆ nrs1∆ double mutants larger than the swi4∆ or swi6∆ single mutants? 

* … and mbp1∆ nrs1∆ and mbp1∆ OPnrs1?

3. Page 10, Figure 5B. The rescue of swi4-ts mbp1∆ by GAL1-NRS1 is very weak. Could this be tested by PCR?

4. Fig. S4B, why swi4ts cells grow worst then swi4ts mbp1∆?

5. Fig. S1B. After Rapamycin treatment of the Nrs1-Myc tagged strain, the anti-Myc antibody detects two bands. Is the upper band a phosphorylated version or Nrs1? Is Nrs1 regulated by protein phosphorylation?

6. Fig. S3C. Are there two bands of Whi5-HA?

Minor points:

Page 3, Introduction, 2nd paragraph: "spindle body"should be "spindle pole body".

Page 6, 1st paragraph: "Figure S1D" should be "Figure S1D and S1E". Figure S1E is not mentioned in the manuscript.

Figure 5B: use the same panel arrangement as in Figure S4B (glucose, raffinose and galactose).

Figure 5C: indicate that the left panels correspond to Glucose and the right panels to Galactose.

Figure legend to Figure S3B: "6 h induction", remove "induction".

Figure legend to Figure S4A: with "plasmids" expressing.

Reviewer #4: In this manuscript, Tollis, Singh et al. identify a recently evolved microprotein of budding yeast, Ylr053c/Nrs1, as a dosage suppressor of G1 arrest of cln3∆ bck2∆ double mutant cells, and investigate its contribution to size control at START in nitrogen-limited medium.

In budding yeast cell size is controlled in a late G1 cell cycle commitment point called START. When the adequate cell size is reached, cells activate the G1/S transcriptional program by antagonizing the Whi5 inhibitor of SBF-driven transcription and by activating the G1-CDK positive feedback loop that pushes cells towards S phase. The most upstream G1 cyclin Cln3 and a poorly characterized Bck2 together contribute to the initial inactivation of Whi5. Strikingly, cell size at START is modulated by growth conditions, with cells being larger in nutrient-rich medium than in nutrient-limited medium. 

How nutrients contribute to Start control has been studied for decades, but some players still seem to be missing. Indeed, while cln3∆ bck2∆ cells are unable to pass Start, deletion of WHI5 bypasses this arrest. Strikingly, triple mutant cells maintain nutrient-dependent size control, suggesting that another pathway contributes to size control in the absence of Whi5. 

In order to identify new players involved in this mechanism, Tollis et al. conducted an elegant yeast overexpression screen looking for dominant suppressors of cln3 bck2 G1 arrest. Doing so, they identified a poorly characterized ORF, which they named NRS1. This new protein falls in the category of microproteins, which are defined as rapidly evolving proteins that give plasticity to cells to adapt quickly to changing environment. The authors use a wide range of well-designed and beautifully-performed yeast genetic experiments, quantitative microscopy, and in vivo or in vitro biochemical approaches to propose a molecular model in which Nrs1 counteracts the Whi5 inhibitory role to promote SBF gene transcription and START in nitrogen-limited conditions. This conclusion is substantiated by showing that i) Nrs1 is strongly induced in nitrogen-limited media or upon TORC1 inhibition by rapamycin; ii) GAL1-NRS1 lowers the cell size of WT, cln3∆ and bck2∆ cells, but not of whi5∆, swi4-ts or swi6∆ cells; iii) Nrs1 interacts physically with Whi5, Swi4 and Swi6; iv) Nrs1 promotes transactivation of SBF-driven genes by counteracting the repression by Whi5.

While there is enough novelty to consider this manuscript for publication in PLoS Biology, it is noteworthy to mention that the authors failed to identify a pathway acting in parallel to the Cln3-Bck2-Whi5 module as initially sought, and that effects on cell size are seen only upon Nrs1 overexpression, not loss of function in any of the conditions they tested. This is honestly acknowledged by the authors. It thus appears that Nrs1's presence boosts SBF transcription by binding and counteracting Whi5 inhibition in nitrogen-limited media, but that its absence does not prevent Start to occur at the right size. 

The following points deserve to be addressed before publication:

Major points:

1. Does GAL-NRS1 also suppress the G1 arrest of cln1 cln2 cln3 triple mutant? This may give insights to the role of Bck2.

2. Nrs1 appears as a doublet on western blots, but as a single band upon TORC1 inhibition by rapamycin. It would be useful to know if this corresponds to a PTM or cleavage of Nrs1. MW markers should be added to estimate the size of this band. Which of the two forms, short or long, does interact with Whi5 and Swi4,5? Also, why are IPs used instead of whole cell extracts? Can Nrs1 not be detected on the latter, why? 

3. The cell cycle regulation of Nrs1 abundance, its dynamics of induction after nitrogen limitation, and nuclear localization should be better documented. It is expected that a regulator of Start would peak just before the G1/S transition, which could be shown by western blots of synchronized or elutriated cells. It is not so clear from the images provided that Nrs1 is nuclear in late G1 unbudded cells. A few examples of higher resolution images and quantification of the fraction of G1 cells with nuclear Nrs1 would be useful.

4. It is not known if Nrs1 also boosts MBF-driven transcription. Has this been tested or is it not relevant? A comment would be welcome.

5. The initial SGA screen identified 12 high-confidence hits. It is shown that YEA4 was not confirmed in a direct test. What about the other hits?

6. Size control by low nitrogen or poor carbon sources is used interchangeably in the text. It would be useful to better distinguish the two controls in the introduction and be more precise in figure legends on which conditions have been used, and for how long.

Minor points:

1. two different concentrations of rapamycin are used in the manuscript. 100 nM for the experiments shown on figures 2 and 200 ng/mL for those on the figure S1, figure 4C. Is there a biological reason for using these 2 different concentrations? Moreover, the length of the rapamycin treatment applied before cell imaging (figure 2A) or protein extracts preparation (figures 2B, S1E, 4A) should be given.

2. The meaning of the dN/dS ratio and how it is calculated would help non-specialists of evolutionary biology if details were given in the material and methods or in supplemental methods.

3. Figure S1B: To support the claim that Nrs1 is transiently expressed, a loading control should be added to this figure. Additionally, an input of a better quality would be appreciated.

4. Figure 4A: To support their model, the authors should show if Nrs1 and Swi4 or Swi6 interact when cells are grown in nitrogen-limited source medium.

5. Figures 4B and S3D: The origin of the recombinant GSTNrs1 fusion protein is not described. Is it expressed from the locus or from a plasmid? The information should be provided in the strain or plasmid table.

6. Figures 4C and S3A: It is reasonable to think that ChIP-PCR experiments were performed more than once. Error bars should be added.

7. Figure S3C: It is not clear why 2 polypeptides that migrate differently are referred to as Whi5HA. Is the faint upper band hyperphosphorylated Whi5?

8. Figure 4D: The authors should indicate if they determined the diffusion constant of the total or nuclear Swi6. If they determine the latter, they should do the same for Nrs1.

9. Figures S5A: An immunoblot is provided to show that the construct Whi5-Nrs1-GFP had the expected molecular weight. It would more convincing if the molecular weight of the ladder was added. Additionally, the difference of expression between Whi5-GFP and Whi5-Nrs1-GFP is really striking. However, protein extracts are prepared from cells grown in nitrogen-limited medium. It is a condition that is never used with cells expressing this construct. An immunoblot performed with protein extracts prepared from cells grown in SC + Glu 2% would be more appropriate and less misleading.

10. Figures S1E, S4A do not appear to be cited anywhere in the text.

11. The type of media (carbon sources) is missing on the panels of the figure5c.

12. A typo on p11 line 2: Figures 6A; Figure S5B…

---

## [Decision Letter · Decision Letter 2]

5 Aug 2021

Dear Dr Tollis,

Thank you for submitting your revised Research Article entitled "A nitrogen source-regulated microprotein confers an alternative mechanism of G1/S transcriptional activation in budding yeast" for publication in PLOS Biology. Please accept my apologies for the delay in getting back to you on your revised manuscript. I have now obtained advice from the original reviewers and have discussed their comments with the Academic Editor. 

Based on the reviews, we will probably accept this manuscript for publication, provided you address the remaining comment from Reviewer #1 (to move Figures S2F/G to the main Figure panels) and the following data and other policy-related requests that I have provided below:

A) We would to suggest the following modification to the title, to make it more compelling for our broad readership:

'The microprotein NRS1 rewires the G1/S transcriptional machinery to enable cell cycle progression during nitrogen limitation in budding yeast'

B) You may be aware of the PLOS Data Policy, which requires that all data be made available without restriction: http://journals.plos.org/plosbiology/s/data-availability. For more information, please also see this editorial: http://dx.doi.org/10.1371/journal.pbio.1001797

Regardless of the method selected, please ensure that you provide the individual numerical values that underlie the summary data for the following Figures, as they are essential for readers to assess your analysis and to reproduce it:

Fig 2C, 2D, 3A-E, 5A-B, 6B, 7-C, Fig S2A-E, S4A-B, S5A-B, S6C-F, S7A

C) Please ensure that your Data Statement in the submission system accurately describes where your data can be found and is in final format, as it will be published as written there. Please ensure that the mass spectrometry data that is currently deposited at the ProteomeXchange Consortium (accession number PXD018681) is made publicly available at this stage, as it is currently on hold. 

D) In addition, please provide the accession number for the RNA-seq data currently deposited at the GEO in your Data Availability Statement, and make sure that the data is made publicly available. 

E) Please also ensure that each of the relevant figure legends in your manuscript include information on *WHERE THE UNDERLYING DATA CAN BE FOUND*, and ensure your supplemental data file/s has a legend

F) We require the original, uncropped and minimally adjusted images supporting all blot and gel results reported in the following Figures:

Figure 5A-C, 7A, 7C, Fig S2C-D, S6, S7B, S7D, S10A-B, S10E, S10G-H

We will require these files before a manuscript can be accepted so please prepare and upload them now. Please carefully read our guidelines for how to prepare and upload this data: https://journals.plos.org/plosbiology/s/figures#loc-blot-and-gel-reporting-requirements.

We expect to receive your revised manuscript within two weeks. 

*Published Peer Review History*

*Early Version*

Sincerely,

Richard

Richard Hodge, PhD

Associate Editor, PLOS Biology

rhodge@plos.org

Reviewer remarks:

Reviewer #1: The authors have adequately addressed my concerns and provided new data regarding the growth defect of nrs1 in nitrogen-poor media. However, these important new findings should be in the paper, not in the supplement (figures S2F and S2G). Otherwise it is ready for publication.

Reviewer #2: The new experiments, in the prototrophic background and in the double mutant with anothe SBF component, support a physiological role. The authors did a good job on the revisions, and I support publication of the manuscript.

Reviewer #3 (Sergio Moreno, signs his review): The authors has answered satisfactorily most of my criticisms and have performed additional experiments to support their conclusions. Therefore, I am happy to accept this revised version of the manuscript for publication in PLOS BIOLOGY.

---

## [Editor Report · Decision Letter 3]

8 Oct 2021

Dear Dr Tollis,

Thank you for submitting your revised manuscript and for your patience as I discussed the revision and the new blot data with the academic editor handling your submission. Thank you very much for letting us know about this issue with the Western blot presented in Figure 5A. 

I have provided some comments from the academic editor below my signature. The outcome of our discussions is that we do have some concerns regarding the strength of the new blot to support the conclusions, since the new data does not demonstrate a reciprocal immunoprecipitation in cells. However, we note that the data in Figure 5B does support an interaction between NRS1 and Swi4/Swi6 using purified proteins. 

(1) At this stage, we would like to issue another round of minor revision and ask that you please provide all of the replicate data that you have for Figure 5A and place it in the Supplementary Information. In addition, we ask that you please replace the current Figure 5A with a blot that provides stronger support for the original claims if possible, and move the current Figure 5A to the supplement. 

(2) Thank you for your explanation in the cover letter for opting to not change the title of the manuscript. We appreciate the reasoning for this and accept that this could be seen as an overstatement. We would like to suggest a second version of the title that I have pasted below:

'The microprotein NRS1 rewires the G1/S transcriptional machinery during nitrogen limitation in budding yeast'

We look forward to receiving your manuscript and please do not hesitate to contact me should you have any questions.

Kind regards,

Richard

Richard Hodge, PhD

Associate Editor, PLOS Biology

rhodge@plos.org

**Academic Editor Comments

The data is not as strong as what was claimed initially, as they can now only co-precipitate Swi4 and Swi6 with Nrs1, and not the other way round. The migration patterns of Swi4 and Swi6 are also a little strange in the IPs, forming strong smears not visible in other blots (see input panel in 5A or Fig S4C or the previous figure 4A). I also do not get what is the pale background signal in the myc blots of the Swi4 and Swi6 IPs. So, from this alone, I would not be very convinced of the claim.

On one hand, it is good that they noticed the problem now and the in vitro interaction presented in 5B confirms the interaction, so the general conclusion stands. On the other hand, this is really a sub-optimal blot.

---

## [Editor Report · Decision Letter 4]

17 Dec 2021

Dear Sylvain,

Thank you for submitting your revised Research Article entitled "A nitrogen source-regulated microprotein confers an alternative mechanism of G1/S transcriptional activation in budding yeast" for publication in PLOS Biology. I have now obtained advice from the Academic Editor handling your submission about the additional immunoprecipitation data provided in Figure 5A.

Whist the new immunoprecipitation blot shows interaction in the double tagged strain, we note the blot does not contain a negative control in the form of an anti-FLAG IP from the Nrs1-Myc single tagged strain, to ensure that Nrs1-myc does not bind non-specifically to the anti-FLAG beads. We note that such a negative control was included in the previous anti-myc IP (now Figure S3C). Therefore, we have concerns that the result presented in Figure 5A is not sufficiently conclusive. We would encourage you to provide a blot that is adequately controlled, as the paper will be ultimately stronger and this will avoid any potential problems post-publication. 

(1) Nevertheless, we ask that you please provide additional discussions that qualify what can be concluded from these results, noting the lack of controls, the supportive data in Figure S3C and the observed in vitro interaction in Figure 5B. In the results section, we note that you have already referenced the polyclonal blot in Figure S3C, but we feel it is important to caveat the lack of negative controls in the manuscript text before we can accept your manuscript (in the absence of a new controlled blot).

(2) In addition, we would like to suggest a modification to the title, to make it more accessible for our broad readership:

"The microprotein NRS1 rewires the G1/S transcriptional machinery during nitrogen limitation in budding yeast"

If you feel that this title is an overstatement, then we would also suggest the following:

"The microprotein NRS1 confers an alternative mechanism of G1/S transcriptional activation during nitrogen limitation in budding yeast"

We look forward to receiving your manuscript and please do not hesitate to contact me should you have any questions. I hope you understand the reasons behind this decision.

Sincerely,

Richard

Richard Hodge, PhD

Associate Editor, PLOS Biology

rhodge@plos.org

PLOS

---

## [Editor Report · Decision Letter 5]

19 Jan 2022

Dear Sylvain,

On behalf of my colleagues and the Academic Editor, Sophie Martin, I am pleased to say that we can in principle accept your Research Article "The microprotein NRS1 rewires the G1/S transcriptional machinery during nitrogen limitation in budding yeast" for publication in PLOS Biology, provided you address any remaining formatting and reporting issues. These will be detailed in an email that will follow this letter and that you will usually receive within 2-3 business days, during which time no action is required from you. Please note that we will not be able to formally accept your manuscript and schedule it for publication until you have any requested changes.

We note that you omitted to change the title, as previously requested, so we have taken the liberty of changing it on your behalf to 'The microprotein NRS1 rewires the G1/S transcriptional machinery during nitrogen limitation in budding yeast’. During the proofs, we ask that you please change the manuscript title in the Supporting Information file in the File Inventory as well (Tollis Singh et al_SI.pdf). 

PRESS

Sincerely, 

Richard

Richard Hodge, PhD

Associate Editor, PLOS Biology

rhodge@plos.org

PLOS
